# Harnessing multi-source hydro-meteorological data for flood modelling in a partially glacierized Himalayan basin

Domenico De Santis<sup>1</sup>, Silvia Barbetta<sup>2</sup>, Sumit Sen<sup>3</sup>, Viviana Maggioni<sup>4</sup>, Farhad Bahmanpouri<sup>2</sup>, Ashutosh Sharma<sup>3</sup>, Ankit Agarwal<sup>3</sup>, Sagar Gupta<sup>3</sup>, Francesco Avanzi<sup>5</sup>, and Christian Massari<sup>2</sup>

- <sup>1</sup>Research Institute for Geo-Hydrological Protection, National Research Council, Rende, 87036, Italy
  - <sup>2</sup>Research Institute for Geo-Hydrological Protection, National Research Council, Perugia, 06128, Italy
  - <sup>3</sup>Department of Hydrology, Indian Institute of Technology Roorkee, Roorkee-247667, Uttarakhand, India
  - <sup>4</sup>Department of Civil, Environmental & Infrastructure Engineering, George Mason University, Fairfax, VA 22030, USA <sup>5</sup>CIMA Research Foundation, Savona, 17100, Italy
  - Correspondence to: Domenico De Santis (domenico.desantis@cnr.it)

Abstract. The southern rim of the Indian Himalayas is highly susceptible to floods during the summer monsoon, making accurate streamflow modelling critical yet difficult due to complex terrain, climate variability, and sparse ground observations. This study uses a conceptual, semi-distributed hydrological model – enhanced with both static and dynamic glacier modules – to reproduce streamflow into the Alaknanda River at Rudraprayag gauge (~8600 km²), a representative basin in northern India. The model was calibrated using multi-variable data, including satellite-based glacier water loss and actual evapotranspiration, also to address bias in the precipitation input. Despite inherent data uncertainties and simplified process conceptualization, the tailored hydrological modelling captured key features of observed streamflow and produced internally consistent water balance estimates. Multi-variable calibration improved the simulation of hydrological fluxes and highlighted the value of using complementary satellite-based information in data-poor mountain regions. Parsimonious precipitation adjustment approaches are proven effective for hydrological applications. However, input data errors such as unaccounted-for heavy precipitation events limited short-term streamflow prediction accuracy. The study demonstrates that a viable, parsimonious modelling strategy can still be developed for data-scarce, monsoon-dominated Himalayan basins, offering insights into the spatiotemporal dynamics of streamflow generating processes, the inter-seasonal redistribution of precipitation, the role of cryosphere contributions, and flood simulation. The approach is transferable to other monsoon-dominated, glacier-influenced, and data-limited mountain catchments facing increasing hydroclimatic risks.

#### 1 Introduction

Advancing knowledge of hydrological processes in the Himalayas is essential because of the regional inherent vulnerability to water-induced hazards, complex dynamics, and lack or shortage of data. In the southern rim of the Indian Himalayas, basins characterized by high mountains and densely populated valleys are prone to flash floods and other river flow-related disasters, due to heavy rains triggered by the interaction between the complex orography and the Indian Summer Monsoon (ISM) (e.g., Kumar et al., 2018a; Dimri et al., 2016, 2021). In these basins, a complex interplay of meteorological,

topographical, and runoff generation factors controls streamflow variability, whose accurate forecast is pivotal for effective flood risk management. However, the peculiarity of the Himalayan region poses significant challenges to understanding and simulating streamflow response.

Investigating the dynamics of hydrological variables in the Himalayan basins is limited by their high spatiotemporal heterogeneity and by the lack of ground-based observations. Not only does the availability of measurements decrease dramatically with altitude and topographic complexity, but gauge precipitation data are also often underestimated due to the wind-induced undercatch of snowfall (e.g., Adam and Lettenmaier, 2003). Furthermore, precipitation products from meteorological models or remote sensing observations also suffer from large uncertainties in complex mountainous terrain (e.g., Azam et al., 2021; Dimri et al., 2021; Girotto et al., 2024; Azimi et al., 2025). In Himalayan basins, several studies have highlighted an underestimation of precipitation data when compared with observed streamflow volume (e.g., Immerzeel et al., 2012; Savéan et al., 2015; Dahri et al., 2016; Li et al., 2020; Ghatak et al., 2018; Saddique et al., 2022). Furthermore, available precipitation datasets are lacking in detecting and characterizing highly localized and short occurrence events such as cloudbursts (Dimri et al., 2016, 2017), which are as impactful in terms of flash flooding as elusive both for sparse ground stations and for remote sensors and meteorological models at coarse scale (e.g., Jena et al., 2020; Garg et al., 2023). Finally, glacier melting also contributes to the streamflow generation, although in monsoon climates it remains a minor component (e.g., Collins et al., 2013; Nie et al., 2021), localized in the season of high flows. Accurate quantification of the ice melt is hampered by the difficulty of monitoring changes in glacier mass. As very few and sparse glaciological field measurements are available, remotely sensed multi-vear difference in glacier surface elevation offers a valid alternative for studying ice storage variations (e.g., La Frenierre and Mark, 2014; Dorigo et al., 2021). More generally, 50 using independent datasets (including satellite-derived ones) for additional hydrological variables can provide valuable information in this high-altitude, data-poor region, but it remains difficult to close the water balance at the basin scale due to inconsistencies with precipitation (e.g., Shafeeque et al., 2019; Li et al., 2020; Miao et al., 2024).

Process-oriented hydrological modelling can bridge this gap, taking advantage of multiple data sources to address uncertainties in model forcings and provide a consistent representation of basin behaviour. Conceptual and semi-distributed models can be particularly well-suited for basins with complex dynamics and limited data availability, resulting comparable to more advanced models for specific application purposes (e.g., Kobierska et al., 2013; Addor et al., 2014; Ragettli et al., 2014; Orth et al., 2015; Finger et al., 2015). Under sub-optimal conditions of data scarcity and limited understanding of the system, highly complex models may indeed provide performance that is not necessarily improved, as well as insights based on inadequate assumptions, resulting in less robustness than simple but efficient models designed to represent the main hydrological processes, with a degree of sophistication functional for targeted modelling (e.g., Burlando et al., 2002; Orth et al., 2015; Horton et al., 2022). In this perspective, model selection or development should be 'fit-for-purpose' (Beven and Young, 2013), looking for the most appropriate structure and complexity in relation to application requirements, hydrological system characteristics, and data availability (e.g., Efstratiadis and Koutsoyiannis, 2010; Fenicia et al., 2011).

Low-complexity models are commonly used for flood simulation in large basins, implemented at appropriately short temporal resolution, including hourly time-step (e.g., Viviroli et al., 2009; Horton et al., 2022), and at low spatial resolution, assuming that the downstream impact of any inadequate capture of small-scale processes can be sufficiently limited (e.g., Momblanch et al., 2019; Wang et al., 2024). For example, advanced methods for glacier-related hydrologic processes are typically applied to small basins (e.g., Naz et al., 2014; Pesci et al., 2023), while at larger scales a simplified representation generally prevails (e.g., Lutz et al. 2013; Yang et al., 2025). Then, hydrological modelling over large domains must properly address the spatial and temporal heterogeneity of processes that influence streamflow generation (e.g., Horton et al., 2022, van Tiel et al., 2023), representing different contributions with a level of accuracy also informed by the basin size (Momblanch et al., 2019). The basins of the southern rim of the Indian Himalayas are monsoon-dominated in the high flow regime (e.g., Thayyen and Gergan, 2010; Lutz et al., 2014; Dimri et al., 2021), with relative meltwater contributions diminishing with decreasing altitude (e.g., Singh et al., 2016a; Wang et al., 2024; Yang et al., 2025). While parsimonious approaches are generally suitable due to the limited availability and quality of data at relevant scales, the complexity of the system representation should be adjusted to consider how rainfall, snow and ice melt contribute differently to the downstream river flow regime. Therefore, in medium to large basins encompassing high mountain and valley environments, with rainfall dominating over snowfall as source of precipitation and low proportion of glacier area, flood-oriented modelling can properly emphasize the representation of rainfall-runoff transformation, simplifying that of glacier and snow dynamics in the upper reaches.

Furthermore, to address data scarcity and limit sensitivity to input errors, it may be particularly useful to enhance the calibration strategy of conceptual models by using multiple reference variables, complementary to conventional streamflow measurements (e.g., La Frenierre and Mark, 2014; van Tiel et al., 2020), rather than increase the model complexity (e.g., Finger et al., 2015; Tarasova et al., 2016). Constraining the hydrological model with additional data can also improve the reliability of calibrated parameters and limit the model's tendency to compensate for errors (e.g., an underestimation of high-altitude precipitation input being compensated by an overestimation of modelled glacier melt). However, a more realistic representation of internal processes does not necessarily lead to better streamflow performance (e.g., Mayr et al., 2013; Finger et al., 2015; van Tiel et al., 2020).

In this perception, the objective of this study is to develop and evaluate a parsimonious, semi-distributed hydrological modelling approach for simulating streamflow under high flow regime in the data-scarce, monsoon-dominated basins in the Indian Himalayas. The approach is based on the one hand on the integration, within a conceptual model well-suited for rainfall-induced floods, of tailor-made snow and glacier modules with low complexity and minimal data requirements. On the other hand, a multi-variable calibration is considered, involving supplementary satellite-based data on actual evapotranspiration and glacier mass loss, to improve the representation of the water balance. The study also explores the impact of precipitation input uncertainties by comparing different adjustment approaches and evaluates the contribution of ice melt in streamflow generation using simplified glacier representations. The modelling approach was tested in the Alaknanda River basin in northern India, where the ISM causes frequent flash floods, sometimes with disastrous effects

(e.g., Joshi and Kumar, 2006; Rautela et al., 2023a, 2023b). Although this study is somewhat preliminary due to the scarcity 100 and high uncertainty of the data, the findings offer practical insights into the challenges of modelling hydrological fluxes in high-altitude, monsoon-dominated basins with glacierized headwaters, and contribute to the understanding of the flood generating processes in similar Himalayan environments.

#### 2 Data and methods

125

### 2.1 Study basin and streamflow gauge data

The Alaknanda River basin is located in the Uttarakhand state in northern India. The study area coincides with the sub-basin subtended by the Rudraprayag gauge, upstream the confluence with the Mandakini River. Further downstream, the Alaknanda River joins the Bhagirathi River forming the Ganges River. The study basin lies between 78°58' to 80°15' E and 29°59' to 31°05' N and extends over an area of 8602 km<sup>2</sup>. It is characterized by heterogeneity in topography, climate, and runoff production mechanisms (Yadav et al., 2020). The elevation range is extremely wide (i.e., from 600 to 7700 m above sea level, approximately), with the upper portions of the basin being dominated by snow and glaciers and lying under steep 110 slopes, which reduce gradually in the lower parts. In the high mountain areas, a significant portion of precipitation occurs as snowfall during the late winter and spring months due to western disturbances, while at lower altitudes most of the precipitation is associated with the ISM. Consequently, streamflow is highly seasonal, being governed by rainfall during monsoon months from June to September and influenced by snow melt in the pre-monsoon season (Chakrapani and Saini, 115 2009).

Streamflow data was collected daily at regular times by the Central Water Commission (CWC). The study period includes 2001-2020 water years (June-May). Digital elevation model data (NASADEM at 30-m resolution) were used to delineate the main drainage network and 19 sub-basins (Fig. 1), with areas ranging between 160 and 778 km<sup>2</sup>.

The Alaknanda River basin has faced devastating floods in last decades, resulting in severe loss of life and extensive damages (e.g., Joshi and Kumar, 2006; Rana et al., 2013). Repeated flash floods are caused by heavy rainfall events, even 120 localized and short-lived (i.e., a few hours), including cloudbursts (Dimri et al., 2016, 2017; Kumar et al., 2018a; Singh and Kansal, 2022), which occur frequently in the monsoon months at altitudes of 1000–2000 m (Mishra et al., 2022; Singh et al., 2023a). The basin is also prone to flash floods triggered or exacerbated by other local events. For instance, on 15-17 June 2013 very severe rainfall resulted in devastating flash floods, landslides, and debris flows, but most fatalities and destruction were caused by the Chorabari glacial lake outburst above the village of Kedarnath, which led to catastrophic flooding downstream (e.g., Allen et al., 2016; Mehta et al., 2016).

Figure 1: Study basin: river network and topography are shown along with the location of glaciers.

#### 130 2.2 ERA5-Land reanalysis data

The ERA5-Land reanalysis dataset was used as meteorological forcings (Muñoz-Sabater et al., 2021). ERA5-Land provides hourly information of land surface variables, available globally at  $0.1^{\circ}$  grid resolution, consistent with atmospheric fields from the ERA5 climate reanalysis. Besides the commonly used meteorological inputs (i.e., precipitation P and air temperature T), other variables were considered to estimate the reference evapotranspiration ET $_0$  with the FAO's Penman-Monteith formulation (Allen et al., 1998). Hourly ET $_0$  was computed following the procedure in Singer et al. (2021) and using separate values of the denominator constant for daytime and nighttime (Allen et al., 2006). The spatial patterns shown in Fig. S2-S4 in the Supplement describe the highly heterogeneous climatic conditions within the basin.

The ERA5-Land dataset was shown to capture precipitation patterns adequately in the Himalayas (e.g., Chen et al., 2021; Kumar et al., 2021; Khadka et al., 2022; Singh et al., 2025). However, difficulties in reproducing processes at fine spatial and temporal scales were highlighted (Khadka et al., 2022; Singh et al., 2025), together with an effect of elevation and season on its performance (Paul et al., 2024; Singh et al., 2025). A wet bias was found during the monsoon by Chen et al. (2021) and Khadka et al. (2022), while according to Kumar et al. (2021) ERA5-Land showed a low bias in tracking large storms (i.e., longer than 5 days).

## 2.3 GLEAM actual evapotranspiration data

Actual evapotranspiration (AET) was obtained from the GLEAM dataset v3.8a at 0.25° spatial resolution (Miralles et al., 2011; Martens et al., 2017). Potential evapotranspiration (PET) was computed with a Priestley-Taylor equation and then

170

175

converted into AET considering an evaporative stress factor. In ice- and snow-covered regions, a Priestley-Taylor equation adapted for ice and super-cooled waters was considered. GLEAM algorithm employs several forcing datasets, such as reanalysis radiation and air temperature, a combination of gauge-based, reanalysis and satellite-based precipitation, and satellite-based vegetation optical depth, as well as it assimilates satellite-based surface soil moisture.

## 2.4 Glacier outlines and stored water loss

Glacier outlines dating back to approximately the year 2000 were obtained from the global Randolph Glacier Inventory, RGI (RGI 7.0 Consortium, 2023). The glaciers have an area of 1042 km<sup>2</sup>, corresponding to about 12% of the total area of the basin, with some sub-basins exceeding 30% (Fig. 1).

Changes in glacier volume were analysed in several local studies (e.g., Remya et al., 2022; Mishra et al., 2023; Bhambri et al., 2023; Bhattacharya et al., 2023; Singh and Pandey, 2024), generally reporting a negative mass balance in recent decades. For this work, the study by Bandyopadhyay et al. (2019) was taken as a reference. Geodetic glacier mass balance data over the period 2000-2014 were calculated from the elevation changes, evaluated on RGI outlines through multi-annual high-resolution satellite-based digital elevation models. The estimated mass balance was validated using reported observations on select glaciers. Summary data at the river basin scale were provided for the two main tributaries (Dhauliganga and Pindar) and the Upper Alaknanda (upstream of the confluence with Dhauliganga). Here, glacier stored water loss estimates were used as independent reference in the model calibration, for the corresponding three groups of sub-basins. To be consistent with Bandyopadhyay et al. (2019), an ice density of 850 kg m<sup>-3</sup> was assumed, appropriate for converting geodetic glacier volume changes (Huss, 2013).

# 165 **2.5 Hydrological model**

The conceptual and semi-distributed MISDc-2L model was used (Massari et al., 2018), already applied in different versions to a multitude of basins with heterogeneous characteristics (e.g., Brocca et al., 2011, 2012; Masseroni et al., 2017; Camici et al., 2020; Nguyen et al., 2020; De Santis et al., 2021). The model was modified here with a tailored snow module and the addition of a static and a dynamic glacier module, integrated for the first time in its structure. Since the study basin is monsoon-dominated in the high flow regime, a simplified conceptualization of the snow and glacier contribution to streamflow was considered, adopting commonly used empirical and parsimonious methods with meteorological forcings at a coarse resolution (e.g., Li et al., 2014; Pohl et al., 2015; Su et al., 2016; Chen et al., 2017a; Ghatak et al., 2018; Chawla and Mujumdar, 2020; Huang et al., 2022; Yang et al., 2022; Nazeer et al., 2022; Laha et al., 2023).

The model was applied at hourly time-step and sub-basin scale. MISDc-2L schematizes the soil in two storage layers, within a soil water balance module that generates surface and subsurface runoff. While these modules were implemented in a lumped way, the input processing was performed at a higher spatial resolution and then averaged across the sub-basin. Specifically, rainfall-snowfall separation, snowpack evolution, snow and ice melting were simulated at every ERA5-Land grid point. Ice melting was simulated only on grid points classified as having afferent glaciers. At sub-basin scale, the

glacier-sourced meltwater flux, averaged on the classified grid points, was multiplied by the glacierized fraction of sub-basin area. The main features of the snow and glacier modules are explained below, while in the Section S1 of the Supplement further details are provided on the hydrological model and its implementation, which includes other changes mainly related to the parameterization compared to the previous formulations.

## 2.5.1 Snow and glacier modules

An air temperature threshold was used for rainfall-snowfall partitioning and snow melting, with the snowpack acting as temporary water storage. Snow melting was simulated with the well-known degree-day method according only to hourly air temperature, and similarly the ice melting over glacierized areas:

$$M_{\text{snow/ice}} = \begin{cases} \text{DDF}_{\text{snow/ice}} \cdot (T - T_0)/24 & T > T_0 \\ 0 & T \le T_0 \end{cases}$$
 (1)

where M is the melt rate from snowpack or glaciers [mm h<sup>-1</sup>],  $T_0$  is the threshold temperature [°C], and DDF is the degree-day-factor [mm °C<sup>-1</sup> d<sup>-1</sup>] differentiated for ice- and snow-covered surfaces. DDF<sub>snow</sub> tends to be lower than DDF<sub>ice</sub>, due to higher albedo of snow compared to ice. Here, a linear proportionality was assumed between DDF<sub>snow</sub> and DDF<sub>ice</sub>:

$$DDF_{ice} = k_{ice} \cdot DDF_{snow}, \qquad (2)$$

with  $k_{ice} > 1$ .





The degree-day method exploits the high correlation of temperature with various components of the surface energy balance, which more properly describes the melting processes (Hock, 2005). A spatial variability of the DDF value is to be expected, given topographic effects (e.g., slope and aspect) and other concurring meteorological variables (e.g., radiation and albedo) (Hock, 2005). Here, the dependence of DDF with altitude was considered, due to the latter's wide variability within the basin. DDF is expected to increase with elevation (e.g., Hock, 2003; Ismail et al., 2023), as confirmed in Himalayas by several studies (e.g., Kayastha et al., 2003; Deng and Zhang, 2018). For example, different DDF values were proposed for altitudes below and above 5000 m in Central Himalaya (Kayastha et al., 2020; Khadka et al., 2020). In this study, a more flexible relationship between DDF and elevation *Z* was proposed:

$$DDF(Z) = \left[ \tan^{-1} \left( \frac{Z - Z_{\text{thr}}}{\text{scale}} \right) \right] \left( \frac{DDF_{\text{max}} - DDF_{\text{min}}}{\pi} \right) + \frac{DDF_{\text{max}} + DDF_{\text{min}}}{2} , \tag{3}$$

where DDF<sub>max</sub> and DDF<sub>min</sub> parameters constitute the range bounds,  $Z_{thr}$  [m] is a location parameter at which the average value of the DDF range is obtained, and the scale parameter [m] controls the smoothness of the DDF transition along Z. Equation (3) is hereinafter referred to snow, but due to Eq. (2) holds also for ice. The current version of the model does not consider either sublimation or meltwater refreezing, nor does it distinguish between debris-free and debris-covered glaciers. To counteract overparameterization issues, 4 of the 6 interdependent parameters were set a priori. Specifically,  $T_0$  was assumed to be equal to 0 °C (e.g., Schaefli et al., 2005, 2014),  $Z_{thr}$  and scale were set at 5000 and 50 m, respectively, to







mimic the threshold tested by Kayastha et al. (2020), and finally  $k_{ice}$  was set at 1.3, following field experiments by Singh et al. (2000) in a nearby glacier.  $DDF_{snow,max}$  and  $DDF_{snow,min}$  parameters were obtained through calibration. This specific setup, although limiting in terms of flexibility, considers that the modelling was not constrained in this study by reference snow dynamics data.

Ice melting is assumed to occur once the seasonal snowpack is locally depleted and refers to the fraction of the sub-basin covered by glaciers. Melting dynamics influence the geometry, including the area, and the actual water storage of glaciers, impacting flow generation. In this study, a static glacier module was firstly considered, without simulating changes in area and volume (e.g., Terink et al., 2015; Nepal et al., 2017; Gao et al., 2017; Laha et al., 2023). To deal with the assumption of infinite ice storage (e.g., Schaefli et al., 2005, 2014; Savéan et al., 2015; Pohl et al., 2015), the model was constrained to glacier mass balance data (e.g., Konz and Seibert, 2010; Jost et al., 2012).

To provide a more realistic conceptual representation of the process, a simplified dynamic glacier module was also tested as a variant. Several formulations have been proposed to simulate glacier evolution with relatively simple representations within hydrological models (e.g., Huss et al., 2010; Wortmann et al., 2019). The use of a volume-area (V-A) scaling relationship can be particularly practical since it directly considers estimates of ice melting to reproduce extent changes for a large set of glaciers (e.g., Luo et al., 2013; Lutz et al., 2013; Su et al., 2016; Van Beusekom and Viger, 2016; Valentin et al., 2018; Chen et al., 2018; Cui et al., 2023; Nunchhani et al., 2024). Here, a novel implementation of the V-A scaling relationship was adopted at the sub-basin scale and applied with parameters set to global values from literature. The dynamic glacier module is described in Appendix A.

## 2.6 Datasets consistency analysis and precipitation adjustment

A preliminary data assessment revealed a significant water budget imbalance, with streamflow and AET far exceeding precipitation and glacier melt. Focusing on ERA5-Land precipitation and CWC streamflow data, the runoff-to-precipitation ratio at the annual scale ranges between 0.78 and 1.33, except for water year 2014 when it is slightly greater than 2. This spurious value may be related to the disastrous flood of June 2013, with observed streamflow resulting persistently high even in the months following the event. For this reason, streamflow data in water year 2014 were excluded from the analysis. Similar inconsistencies between precipitation and streamflow data have been described in previous works in Himalayan basins (e.g., Lutz et al., 2014; Savéan et al., 2015; Li et al., 2020; Ghatak et al., 2018; Shafeeque et al., 2019; Saddique et al., 2022; Miao et al., 2024), while Goteti and Famiglietti (2024) attributed the observed imbalance in Indian basins to underestimation in precipitation datasets rather than to change in basin water storage, inter-basin groundwater flow, and anthropogenic influences. Streamflow measurements (together with other water budget terms) are typically assumed as a benchmark, and precipitation is adjusted for biases (e.g., Duethmann et al., 2013, 2015; Lutz et al., 2014; Savéan et al., 2015; Wortmann et al., 2018). Explicit correction parameters for precipitation input are commonly applied in hydrological modelling in high-altitude regions, also to consider bias due to gauge undercatch and limited representativeness, as well as





further implicit adjustment of meteorological forcings occurs through the tuning of altitudinal gradients (e.g., Mayr et al., 2013; Van Beusekom and Viger, 2016; Wang et al., 2021; Ruelland, 2024).

Here, different systematic error structures were assumed and lumped parameters were estimated simultaneously with those of the hydrological model. The proposed approaches are parsimonious but omit the spatial and temporal bias variability typical of precipitation data, as well as they are ineffective in the case of missing (or severely underestimated) events in the meteorological dataset.

First, the use of a multiplicative, time-invariant coefficient CF was considered for adjusting precipitation:

$$P_{\text{adj},1} = \text{CF} \cdot P \tag{4}$$

In a second case  $(P_{adj,2})$ , seasonal CF values were assumed for two different 6-month periods (i.e., from May to October and from November to April), which include summer monsoon and western disturbance systems respectively.

As a third alternative, a two-parameter, time-invariant adjustment formulation was used (e.g., Bannister et al., 2019):

$$P_{\text{adj,3}} = \text{CF}_{\text{COE}} \cdot P^{\text{CF}_{\text{EXP}}} \tag{5}$$

# 2.7 Calibration setup and evaluation strategy

Warmup, calibration, and validation periods cover water years 1999-2000, 2001-2014, and 2015-2020, respectively. The calibration period was set to approximately match the time coverage of the glacier mass balance data. In the comparison between observed and simulated streamflow, 2014 was excluded as explained above.

A multi-variable and multi-response objective function (Efstratiadis & Koutsoyiannis, 2010) was defined to summarizes the features that the model should best fit with respect to the given data (e.g., van Tiel et al., 2020). Specifically, four performance metrics were aggregated into a scalar function, and a single-objective global optimization algorithm was applied, the Covariance Matrix Adaptation Evolution Strategy (Hansen et al., 2003). Despite some disadvantages (Efstratiadis & Koutsoyiannis, 2010), such an embedded multi-criteria calibration approach is widely used (e.g., Gao et al., 2017; van Tiel et al., 2018; Mei et al., 2023), adopting suitable weights for an acceptable trade-off in the simulation of the individual components of interest. In this regard, the practice of weight refinement during optimization tests was followed here (e.g., Viviroli et al., 2009; Tarasova et al., 2016; Sleziak et al., 2020; Ruelland, 2024).

The model calibration was developed considering the following scenarios:

- Scenario 1 (baseline): the model was calibrated against reference streamflow (considering two metrics, one of which is specific for high flows in terms of annual peaks), AET and glacier water loss data simultaneously, adopting the static glacier module and the multiplicative, time-invariant precipitation adjustment (P<sub>adj,1</sub>).
- Scenario 2: same as in 1, but the model was not calibrated for glacier water loss, with the latter not being simulated.
- Scenario 3: same as in 2, but the model was no longer calibrated against reference AET.

different formulation for precipitation adjustment.


- Scenario 4: same as in 3, but only one metric was considered in the calibration against reference streamflow, excluding the specific one for peak flows.
  - Scenario 1B: same as in 1 but considering different coefficients for precipitation adjustment depending on the period of the year  $(P_{adj,2})$ .
  - Scenario 1C: same as in 1 but considering a two-parameter, time-invariant approach for precipitation adjustment  $(P_{\text{adj},3})$ .
  - Scenario 1D: same as in 1 but considering the dynamic glacier module based on a V-A scaling relationship (Appendix A).

In Scenario 1 (and its variants 1B, 1C, and 1D), the overall objective function  $\Phi$  to be minimized therefore involves a weighted combination of four efficiency indices  $\varphi_i$ :

$$\Phi = \sum_{i} w_i (1 - \phi_i) , \qquad (6)$$

$$\phi_1 = \text{KGE} \,, \tag{7}$$

$$\phi_2 = 1 - APFB \,, \tag{8}$$

$$\phi_3 = 1 - \frac{|AET_{sim} - AET_{ref}|}{AET_{ref}}, \tag{9}$$

$$\phi_4 = 1 - \frac{\sum_{j \mid \text{IMV}_{j,\text{sim}} - \text{IMV}_{j,\text{ref}}}}{\sum_{j \mid \text{IMV}_{j,\text{ref}}}}.$$
(10)

KGE is the Kling-Gupta efficiency index (Gupta et al., 2009), computed between observed and simulated streamflow, while APFB is the annual peak flow bias proposed by Mizukami et al. (2019). AET<sub>sim</sub> and AET<sub>ref</sub> indicate the simulated and reference AET volume at basin scale, while IMV<sub>j,sim</sub> and IMV<sub>j,ref</sub> are the simulated and reference glacier-sourced meltwater volume for the *j*-th sub-basin aggregate. In the following, the terms φ<sub>2</sub>, φ<sub>3</sub>, and φ<sub>4</sub> are more intuitively referred to as Eff<sub>APFB</sub>, Eff<sub>AET</sub>, and Eff<sub>IMV</sub>, respectively. The weights vector *w* is equal to [0.65, 0.1, 0.1, 0.15]. In Scenarios 2, 3, and 4, Eff<sub>IMV</sub>, 290 Eff<sub>AET</sub>, and Eff<sub>APFB</sub> were progressively omitted from the objective function during calibration (see Tab. 1). Then, ice melting was not modelled in these alternative scenarios. This is motivated by i) the difficulty in realistically reproducing the process without using specific constraints (mainly due to underestimation of precipitation), and ii) the small glacier-sourced supply in the water budget at basin scale. The same 13 parameters are calibrated in all scenarios, except in 1B and 1C which have a


|                      |                     | Scenario |   |   |   |    |    |    |
|----------------------|---------------------|----------|---|---|---|----|----|----|
|                      |                     | 1        | 2 | 3 | 4 | 1B | 1C | 1D |
| Calibration criteria | KGE                 | X        | X | X | X | X  | X  | Х  |
|                      | Eff <sub>APFB</sub> | X        | X | X |   | X  | X  | X  |
|                      | Eff <sub>AET</sub>  | X        | X |   |   | X  | X  | X  |
|                      | Eff <sub>IMV</sub>  | X        |   |   |   | X  | X  | Х  |
| Glacier module       | Static              | X        |   |   |   | X  | X  |    |
|                      | Dynamic             |          |   |   |   |    |    | X  |
|                      | None                |          | X | X | X |    |    |    |
| Precipitation bias   | $P_{ m adj,1}$      | X        | X | X | X |    |    | Х  |
| adjustment           | $P_{ m adj,2}$      |          |   |   |   | X  |    |    |
|                      | $P_{ m adj,3}$      |          |   |   |   |    | X  |    |

#### 300 Table 1. Scenarios configuration.

Other performance metrics were also considered for the evaluation of modelled streamflow (see Sect. 3.2). Specifically, the three components of KGE index were analysed, i.e., the Pearson correlation coefficient r, and the ratios  $\sigma_{\text{sim}}/\sigma_{\text{obs}}$  and  $\mu_{\text{sim}}/\mu_{\text{obs}}$ , where  $\sigma_{\text{sim}}$  are the standard deviation and the mean of simulated streamflow, while  $\sigma_{\text{obs}}$  and  $\mu_{\text{obs}}$  are those of observed streamflow. Then, the KGE index based on a root squared transformation of streamflow time series (KGE<sub>sqr</sub>) was assumed informative for average flow conditions (Garcia et al., 2017), while the inverse transformed streamflow (KGE<sub>inv</sub>) was considered to emphasize low flows (Santos et al., 2018). In addition, to overcome the effect of streamflow seasonality on the performance metrics (e.g., van Tiel et al., 2020) and give even more relevance to the high flow regime, KGE index was recalculated considering only the monsoon period from June to September (KGE<sub>JJAS</sub>). Finally, the well-known Nash-Sutcliffe efficiency index (NSE) was computed. Furthermore, the internal model behaviour was assessed in terms of reliable spatiotemporal patterns and reasonable representation of the hydrological processes (see Sect. 3.3).

# 3 Results



# 3.1 Calibration and validation analysis

The values of the calibrated parameters in the different scenarios are reported in Tab. S1 in the Supplement, whereas Tab. 2 shows the efficiency indices in the objective function evaluated during the calibration and validation periods.

In the calibration phase, the baseline scenario produced an acceptable KGE of 0.88, which progressively improved up to 0.91 moving along Scenarios 2-4, due to the reduction of the competing criteria. Regarding the baseline variants, the more realistic representation of glacier dynamics in Scenario 1D did not increase the KGE. Conversely, a more flexible adjustment of precipitation was able to improve the streamflow simulation. Specifically, the precipitation adjustment that includes two






seasonal parameters (Scenario 1B) provided a KGE equal to 0.90 and enhanced the glacier melting simulation, whereas the adoption of a more complex structure in Scenario 1C resulted in the largest KGE (0.93), at the expense of the other objective metrics.

In the validation phase, KGE decreased overall, following a similar behaviour between scenarios with some exceptions. Scenario 1D performed better than Scenario 1 and both outperformed Scenario 2. Scenario 1B - and not only 1C - showed higher KGE than Scenarios 3 and 4, despite the greater relative weight of this metric in the latter during calibration. The improvements in streamflow across the different scenarios obtained during the calibration phase but not confirmed in the validation one (Scenarios 2-4) can be indicative of a poor representation of physical processes and hydrological behaviour of the basin when additional data are not considered for parameter tuning.

Furthermore, during calibration, the model captured almost perfectly both the mean annual peak flow and the reference AET volume, if the Eff<sub>APFB</sub> and Eff<sub>AET</sub> metrics were incorporated into the objective function. Otherwise, AET was underestimated, due to a smaller adjustment in precipitation (Scenarios 3 and 4). Eff<sub>APFB</sub> showed a value of 0.94, with an overestimation of mean annual peak flow, when not integrated in the calibration (Scenario 4). In the validation phase, a worsening of Eff<sub>APFB</sub> and, to a less significant extent, of Eff<sub>AET</sub> was generally observed. In Scenario 2, Eff<sub>APFB</sub> was lower than in Scenario 1, despite the greater relative weight during calibration. Scenarios 1B and 1C also performed worse than the baseline, whereas Scenario 1D had a slightly higher Eff<sub>APFB</sub>, resulting lower than Scenario 3 only, where the criterion for peak flows had its maximum relative weight in calibration. In Scenarios 3, 1B, and 1C, the model slightly underestimated the mean annual peak flow during the validation period, whereas in the other scenarios an overestimation was observed.

The efficiency index for the glacier stored water loss, Eff<sub>IMV</sub>, has a higher relative complexity since it incorporates spatially explicit information on 3 sub-basin aggregates, whereas in the model only 2 lumped parameters were calibrated to reproduce snow and ice melt dynamics at different altitudes. Nevertheless, in the scenarios where glacier melting was simulated, efficiencies ranged from 0.982 to 0.995, thus capturing the independently estimated and spatially variable stored water loss during calibration.

Regarding precipitation adjustment, an increase of 29% was achieved in Scenario 1 (and similarly in 1D). In Scenario 2, where the omitted contribution of glacier melt had to be compensated, this correction was slightly higher (32.5%), while in Scenarios 3 and 4 it dropped to just over 20%, no longer being constrained to support the reference AET volume. In Scenario 1B, winter precipitation, which mainly occurs as snowfall due to western disturbances and represents a more significant contribution at high altitudes, increased compared to summer precipitation (53% vs 16%). Scenario 1C enhanced hourly precipitation below approximately 1.4 mm and decreased the higher rates, which are more common in the valley areas during the ISM. Therefore, precipitation adjustments suggest that the underestimation is not primarily related to monsoon rainfall, particularly for the more intense events identified in the coarse scale meteorological dataset.

PET was estimated to be generally close to ET<sub>0</sub> in scenarios having calibration constrained against AET. The exception was Scenario 1B, where the seasonal precipitation adjustment involved a reduced water input in the summer months resulting in lower soil water content, with the latter modulating the AET-to-PET ratio. This translated in a higher evapotranspiration




demand (which is at its maximum in the ISM period) to provide an AET volume equal to the reference. In Scenarios 3 and 4, without constraints, the model underestimated both PET (reduced by approximately 40%) and AET (35%) and its ability to capture this process was limited by the bias in precipitation not corrected sufficiently.

The parameters used in the snow and glacier modules assumed consistent values between the different scenarios in which ice melting was simulated, i.e., Scenario 1 and its variants, with DDF<sub>snow</sub> varying approximately between 2.5 and 3.8 mm °C<sup>-1</sup> d<sup>-1</sup> depending on altitude, according to Eq. (3). In the other scenarios, these values significantly increased (on average from 4.3 to 8 mm °C<sup>-1</sup> d<sup>-1</sup> depending on altitude), with the accelerated snow melt functional to an effective reproduction of the rising limb of the seasonal hydrograph (see Sect. 3.2). Furthermore, the enhancement of melting factors compensated for occurrences of modelled snow accumulation at high altitudes due to a locally low positive degree-day sum.

Finally, the remaining parameters operate in the processes of runoff generation, convolution at sub-basin outlet, and propagation in the downstream network. The complex interactions of these processes make it more difficult to identify parameters across scenarios, which does not exclude the possibility of equifinality issues. However, a particularly fast response time generally emerged in the surface runoff convolution, as shown by the low values of calibrated parameter  $\gamma_{GIUH}$  (Tab. S1), which appears compatible with the river network flashiness due to the steep slopes.

| Period | $\varphi$           | Scenario |       |       |       |       |       |       |  |  |  |
|--------|---------------------|----------|-------|-------|-------|-------|-------|-------|--|--|--|
|        |                     | 1        | 2     | 3     | 4     | 1B    | 1C    | 1D    |  |  |  |
| CAL    | KGE                 | 0.882    | 0.894 | 0.907 | 0.911 | 0.904 | 0.930 | 0.878 |  |  |  |
|        | Eff <sub>APFB</sub> | 1.000    | 1.000 | 1.000 | 0.940 | 1.000 | 0.999 | 1.000 |  |  |  |
|        | Eff <sub>AET</sub>  | 1.000    | 1.000 | 0.651 | 0.642 | 1.000 | 0.999 | 1.000 |  |  |  |
|        | Eff <sub>IMV</sub>  | 0.993    | 0     | 0     | 0     | 0.995 | 0.982 | 0.993 |  |  |  |
| VAL    | KGE                 | 0.828    | 0.823 | 0.856 | 0.860 | 0.873 | 0.918 | 0.831 |  |  |  |
|        | Eff <sub>APFB</sub> | 0.966    | 0.955 | 0.982 | 0.935 | 0.952 | 0.951 | 0.968 |  |  |  |
|        | Eff <sub>AET</sub>  | 0.998    | 0.998 | 0.648 | 0.640 | 0.999 | 1.000 | 0.998 |  |  |  |

Table 2. Efficiency indices considered for the objective function, during the calibration (CAL) and validation (VAL) periods, for the different scenarios. In bold the values of the metrics optimized during the calibration.

## **370 3.2 Streamflow**


Additional performance metrics provided complementary information on the model's ability to reproduce observed streamflow at the outlet (Tab. 3). The simulated streamflow tends to have a larger variability than the observed one in the validation phase, whereas all scenarios maintain a very low bias in terms of mean streamflow. In Scenarios 1B and 1C, the more complex precipitation adjustments better address streamflow bias issues. The omission of the glacier melting process (Scenarios 2-4) coincides with the largest overestimation of the mean streamflow in validation. The slight improvement in KGE in Scenario 1D compared to the baseline is attributed to the two bias terms being close to the target value, also due to







the glacier shrinkage resulting in lower ice melt volume. NSE follows similar patterns to KGE, albeit with lower values. The model does not present criticalities in reproducing the average flow conditions (with KGE<sub>sqr</sub> higher than 0.90 and 0.89 during calibration and validation, respectively), but rather in the monsoon period (KGE<sub>JJAS</sub>) and, to a lesser extent, in the low flow regime (KGE<sub>inv</sub>). Regarding the latter, both modelling choices and calibration strategies were aimed to capture high flow conditions, and despite this, KGE<sub>inv</sub> not lower than 0.77 and 0.81 was achieved during calibration and validation, respectively. Nevertheless, performances during the monsoon are less accurate, with a larger decrease in the validation period than that of the KGE over all seasons. The use of additional data in the calibration framework appears more penalizing on the KGE than in the overall period, while the selection of the precipitation adjustment scheme results to be more effective: for example, in the validation phase, when switching from Scenario 1 to 1C, the efficiency index changes from 0.60 to 0.80.

Such localized shortcomings appear to be attributable to two factors. The first is errors on individual events not well captured in the precipitation dataset, for example during the transition from the pre-monsoon to the monsoon periods (May-June). The second is a delay in the reproduction of the rising and falling limbs of the hydrograph on a seasonal scale, which depends on the modelling scenario. Figure 2 presents observed and simulated streamflow time series for the baseline scenario, for which both issues can be detected, whereas Fig. 3 shows the corresponding empirical cumulative distribution functions and scatterplots. The model provides a quite good streamflow reproduction both for calibration and validation periods (Fig. 2) and adequately captures the distribution of observed values (Fig. 3a and 3c), especially in calibration and in any case reproducing well the maximum ones (as also quantified by the Eff<sub>APFB</sub> index). However, the scatterplots (Fig. 3b and 3d) highlight a concentration of systematic deviations in specific streamflow ranges, attributable to delays in the simulation in the onset and end of the monsoon season. In this regard, the role of modelling setup is shown in Fig. S5 and S6 in the Supplement, for Scenarios 1-4 and for the variants of Scenario 1, respectively. The systematic shift does not emerge significantly in Scenarios 3 and 4, where the calibration was only against streamflow. In Scenario 1, the constrained modelling of glacier melt leads to lower DDF values, as shown in Tab. S1, i.e., snow melting is slowed down to maintain snowpack coverage, whereas the observed streamflow begins to increase seasonally in a way that cannot be supported by simultaneous rainfall inputs alone. Of all the scenarios where glacier melting is modelled, in 1C no temporal shift in streamflow is evident, due to the greater flexibility of the precipitation adjustment scheme in modulating the inflow distribution. In Scenario 2, a residual delay can still be noticed in the simulated streamflow, where supporting an increased AET results in a different partition between surface and subsurface runoff compared to Scenarios 3 and 4. More specifically, there is a lower and delayed production of baseflow, with the latter being more effective in capturing the seasonal cycle in observed streamflow than the surface component. This is due to the model's tendency to maintain water availability in the topsoil storage to meet the higher evapotranspiration demand, which also implies increased surface runoff generation during monsoon rains.

| Period | Metric                            | Scenario |       |       |       |       |       |       |  |  |
|--------|-----------------------------------|----------|-------|-------|-------|-------|-------|-------|--|--|
| renou  |                                   | 1        | 2     | 3     | 4     | 1B    | 1C    | 1D    |  |  |
| CAL    | r                                 | 0.885    | 0.895 | 0.907 | 0.911 | 0.905 | 0.930 | 0.882 |  |  |
|        | $\sigma_{ m sim}/\sigma_{ m obs}$ | 1.021    | 1.011 | 0.996 | 0.996 | 1.002 | 1.001 | 1.015 |  |  |
|        | $\mu_{ m sim}/\mu_{ m obs}$       | 0.987    | 0.990 | 0.999 | 1.002 | 0.987 | 1.006 | 0.975 |  |  |
|        | $KGE_{sqr}$                       | 0.912    | 0.921 | 0.931 | 0.934 | 0.924 | 0.946 | 0.907 |  |  |
|        | KGE <sub>inv</sub>                | 0.780    | 0.783 | 0.792 | 0.783 | 0.788 | 0.872 | 0.777 |  |  |
|        | KGE <sub>JJAS</sub>               | 0.707    | 0.735 | 0.766 | 0.777 | 0.760 | 0.814 | 0.699 |  |  |
|        | NSE                               | 0.765    | 0.788 | 0.815 | 0.823 | 0.809 | 0.860 | 0.759 |  |  |
| VAL    | r                                 | 0.885    | 0.896 | 0.915 | 0.918 | 0.905 | 0.929 | 0.880 |  |  |
|        | $\sigma_{ m sim}/\sigma_{ m obs}$ | 1.127    | 1.142 | 1.113 | 1.109 | 1.083 | 1.041 | 1.119 |  |  |
|        | $\mu_{ m sim}/\mu_{ m obs}$       | 1.015    | 1.026 | 1.027 | 1.027 | 1.003 | 1.001 | 1.000 |  |  |
|        | $KGE_{sqr}$                       | 0.891    | 0.896 | 0.920 | 0.926 | 0.917 | 0.940 | 0.890 |  |  |
|        | KGE <sub>inv</sub>                | 0.815    | 0.830 | 0.841 | 0.832 | 0.830 | 0.892 | 0.810 |  |  |
|        | KGE <sub>JJAS</sub>               | 0.597    | 0.601 | 0.680 | 0.690 | 0.678 | 0.800 | 0.591 |  |  |
|        | NSE                               | 0.724    | 0.742 | 0.797 | 0.805 | 0.787 | 0.850 | 0.718 |  |  |

Table 3. Complementary performance metrics for the analysis of streamflow simulation, during the calibration (CAL) and validation (VAL) periods, for the different scenarios.

Figure 2: Time series of observed (grey) and simulated (red) streamflow obtained for Scenario 1, in calibration (a) and validation (b) periods. Represented data are normalized with respect to the maximum observed streamflow value.


Figure 3: Comparison between observed (green) and simulated (red, dotted) streamflow obtained for Scenario 1, in terms of empirical cumulative distribution function in calibration (a) and validation (c), and scatterplot in calibration (b) and validation (d). Represented data are normalized with respect to the maximum observed streamflow value.




### 3.3 AET, glacier melting, and water budget assessment

The model was able to reproduce AET in terms of overall volume at the basin scale, when reference data are considered in the calibration. Figure 4 presents monthly timeseries of simulated and reference lumped AET for Scenario 1, showing a good agreement with a root mean square error equal to 3.3 mm month<sup>-1</sup>. Figure 5 illustrates spatial distributions of mean annual AET at sub-basin and pixel scales for MISDc-2L and GLEAM data, respectively. AET patterns are consistent, although the reference dataset is characterized by slightly greater variability in magnitude between headwaters and downstream areas compared to the hydrological model. This can be attributed to transitions in land cover from snow- or ice-dominated to vegetated surfaces, which are considered in GLEAM and not in the hydrological model (the latter being based on a lumped parameter to compute PET from ET<sub>0</sub>).

Figure 4: Comparison between simulated (in red) and reference (in blue) monthly AET data at basin scale in the study period for Scenario 1.

Figure 5: Comparison between simulated and reference mean annual AET in the water years 2001-2020, for Scenario 1.







A similar additional analysis was also performed for glacier melt, where reference data are available only in the calibration period and on a spatial scale consisting of three sub-basin aggregates. Table 4 reports reference and simulated glacier water loss data, for Scenarios 1 and 1D. The sub-basins outlined here were aggregated as per Bandyopadhyay et al. (2019), considering in addition also the small glaciers in the middle course of the Alaknanda River, between the confluence with Dhauliganga and that with Pindar rivers. Compared to the reference, the model accurately reproduces water loss volumes, with a non-negligible deviation only in the Pindar River basin, although consistent with the uncertainty value (about 60 Mm³) locally indicated in Bandyopadhyay et al. (2019). This may be partly related to the formulation adopted for Eff<sub>IMV</sub>, which focuses on absolute and not relative errors, and therefore favours accuracy on sub-basin aggregates with higher ice melt volume within lumped parameters calibration. The model calibrated from the geodetic glacier mass change over the period 2000-2014 was used to reconstruct an annual distribution of stored water losses (Fig. 6), with ice melting which overall shows considerable interannual variability. A model-based representation of the spatial patterns of glacier stored water loss was also provided, in terms of cumulative volume over the study period (Fig. 7). Overall, high heterogeneity in the estimated glacier melt contribution to streamflow is observed, due to the variability in snowpack dynamics and temperature conditions, as well as in glaciated areas distribution.

Scenario 1D reproduces the glacier runoff contribution in the period and areas involved in the calibration with spatiotemporal patterns similar to those of the baseline scenario, while elsewhere the dynamic module returns lower glacier loss estimates (Tab. 4). During the validation period, overall changes in glacier extent result in an 8% reduction in meltwater compared to the baseline scenario. At the basin scale over the 2001-2020 period, the simulation provides a reduction in glacier areas of about 62 km², with a variation of the ice-covered fraction equal to 0.7% (from 12.1% to 11.4% of the total basin area). This decrease varies in the different sub-basins, exceeding 1.7% in some cases.

The modelled glacier melting for the validation period was compared with previous studies. Here, for the baseline scenario, the simulated water loss rate at basin scale is higher in the calibration period (614 Mm³ y⁻¹ on average) in comparison to the validation one (580 Mm³ y⁻¹), mainly due to variations in the Upper Alaknanda River basin (from 128 to 102 Mm³ y⁻¹). This can be explained by the difference in temperature conditions in the high-altitude areas of the basin and especially in the north-western part (Fig. S7 in the Supplement). However, according to satellite-based analysis (Bhambri et al., 2023; Bhattacharya et al., 2023), in the same area the rate of glacier mass loss observed in the period 2015–2020 appears to have increased significantly compared to 2000–2015. Similarly, an acceleration in glacier retreat was also detected in the Dhauliganga River basin between 2013-2020 compared to 2000-2013 (Singh and Pandey, 2024). These discrepancies highlight the limitations of a simplified and large-scale modelling of glacier dynamics, which, although functional for hydrological purposes, did not appear in validation to be able to reproduce as accurately the remote sensing observations already captured in calibration.

| Period | Data                | Sub-basin aggregate |             |         |           |  |  |  |
|--------|---------------------|---------------------|-------------|---------|-----------|--|--|--|
|        |                     | Upper               | Dhauliganga | Pindar  | Middle    |  |  |  |
|        |                     | Alaknanda           | Dilaunganga | Tilldal | Alaknanda |  |  |  |
| CAL    | Model - Scenario 1  | 1796                | 5037        | 1017    | 740       |  |  |  |
|        | Model - Scenario 1D | 1795                | 5030        | 1014    | 612       |  |  |  |
|        | Geodetic reference  | 1795                | 5037        | 965     | N/A       |  |  |  |
| VAL    | Model - Scenario 1  | 614                 | 2094        | 461     | 310       |  |  |  |
|        | Model - Scenario 1D | 577                 | 1995        | 436     | 183       |  |  |  |

Table 4. Glacier water loss data in Mm<sup>3</sup>, for Scenarios 1 and 1D.

Figure 6: Annual distribution of simulated glacier water loss, for different sub-basin aggregates according to Scenario 1.





Figure 7: Simulated glacier melting in the water years 2001-2020 at sub-basin scale, attributed to the glacier areas represented therein, for Scenario 1.

Additional information on the spatial patterns of the simulated streamflow is reported in Fig. S8 in the Supplement, showing the highly variable distribution of runoff coefficient and generation at sub-basin scale, with maximum runoff coefficient at mid-altitudes (where streamflow is enhanced by increased ice melt, in conditions of low PET) and minimum streamflow generation at the extreme altitudes (due to low precipitation). Then, Fig. 8 summarises the analysis of the modelled water balance at the basin scale, for the baseline scenario. The following formulation has been considered for the water balance:

$$R + S_f + I_{\text{melt}} - Q_{\text{base}} - Q_{\text{surf}} - AET = \Delta S_{\text{soil}} + \Delta S_{\text{snow}},$$
(11)

where terms include adjusted rainfall (R) and snowfall ( $S_f$ ), ice melting ( $I_{melt}$ ), subsurface ( $Q_{base}$ ) and surface ( $Q_{surf}$ ) runoff, AET, and changes in soil ( $\Delta S_{soil}$ ) and snowpack ( $\Delta S_{snow}$ ) water storages. Net of a certain interannual variability, the study period showed an annual mean of around 1980 mm of adjusted precipitation (through a 29% increase), of which more than 75% as rainfall. Simulations outflows are about 1575 and 435 mm per year on average for streamflow and AET, respectively. The water balance closes with approximately 70 mm of modelled glacier melting and 40 mm of snowpack accumulation (the latter due to some areas in ERA5-Land dataset with low positive degree-day sum). The variation of simulated soil water storage over the study period is almost zero. Finally, Fig. 9 shows the seasonal variability of both



rainfall, snow and ice melt inputs, and AET, subsurface and surface runoff outputs of the hydrological system. It emerges the well-known ISM-driven rainfall and runoff seasonal patterns with peaks in July and August. Furthermore, the model shows that ice melt mainly occurs between July and September with the maximum in August, and snow melt generates the maximum water input in June and July, while in May (before the onset of the monsoon) it provides a contribution approximately equivalent to that of rainfall.

Figure 8: Simulated water balance for the study basin, according to Eq. (11), for Scenario 1.

Figure 9: Monthly distribution of subsurface ( $Q_{base}$ ) and surface ( $Q_{surf}$ ) runoff generation, AET, snow melting ( $S_{melt}$ ), ice melting ( $I_{melt}$ ), and rainfall (R), at basin scale over the study period, for a selection of scenarios (from left to right: 1, 1B, 1C, and 4).

#### 4 Discussion





#### 4.1 Precipitation errors

Errors in hydrological model forcings, particularly in precipitation data, are confirmed as main limiting factors in predictive skills and processes understanding (e.g., Lundquist et al., 2019; Tang et al., 2023; Wang et al., 2024). In the study area, precipitation is originated by different weather systems during the year, whose interactions with the highly rugged topography give rise to complex and difficult to capture generation mechanisms. Although meteorological models are a viable option for hydrological input in complex mountainous terrain (Lundquist et al., 2019), orographic rainfall and snowfall remain difficult to simulate (e.g., Viviroli et al., 2011; Azam et al., 2021), thus also limiting the evaluation of the representation of high-elevation processes in hydrological models. At the same time, the latter may be used to benchmark the quality of meteorological forcings for hydrological applications (e.g., Duethmann et al., 2013; Evin et al., 2024).

Here, precipitation data were adjusted using different parsimonious formulations, with increasing flexibility resulting in improved hydrological performance. Scenario 1C, adopting a two-parameter formulation, showed the best results in terms of streamflow, together with a consistent representation of the underlying processes. Model-derived indications can be drawn regarding possible systematic errors of the ERA5-Land data within the study basin. Simulations suggest that precipitation required an average increase of nearly 30%, with the model tending to attribute this underestimation to winter and/or low intensity data, rather than to the high summer rainfall that mainly affects the valley areas. In this sense, more flexible precipitation adjustments (Scenarios 1B and 1C) as well as calibration based only on streamflow (e.g., Scenario 4) tend to reduce the inflow in the months from July to September compared to the baseline scenario (Fig. 9). Although these corrections may be affected by compensation for model errors, the results are consistent with the ERA5-Land validation analyses described in Sect. 2.2 and with other studies indicating more generally larger precipitation biases in winter and at higher altitudes (e.g., Shafeeque et al., 2019; Pritchard, 2021; Saddique et al., 2022).

While more complex and targeted bias adjustment schemes could be beneficial, such systematic corrections are not effective on heavy rainfall events that were not well detected in the coarse meteorological dataset, and that proved to be detrimental on hydrological prediction in high flow regime. It is noteworthy that these coarse scale errors are generally not corrected by statistical downscaling methods (typically employed in hydrological studies over the more demanding dynamical ones), since they do not consider the non-stationarity of spatial precipitation patterns (e.g., Lundquist et al., 2019), particularly at the storm-specific time scale. Finally, input errors not only impact hydrological performance but also the behaviour of the conceptual model. The latter is expected to compensate for inaccuracies in precipitation (e.g., Magnusson et al., 2011; Duethmann et al., 2013; van Tiel et al., 2020; Tang et al., 2023), erroneously adjusting the parameters and misrepresenting underlying processes. For example, the possibility of missing or underestimated rainfall events during the transition from the pre-monsoon to the monsoon periods was highlighted in Sect. 3.2, that can be compensated by the model through accelerated snow melt (Scenarios 2 to 4, where snowpack dynamics are not constrained against glacier loss data).








## 4.2 Hydrological model skills and value of additional calibration data

The proposed conceptual model, with limited data requirements and complexity and tailored to represent specific hydrologic conditions, was combined with a multi-variable calibration strategy and tested in a representative basin in the Indian Himalayas. The use of non-physical parameters, while on the one hand can increase the flexibility in simulating target outputs, on the other hand can hinder reproduction of the internal processes. In this sense, the model was expected to compensate for errors, simulating the streamflow dynamics well but without implicitly capturing other fluxes, unless they are used as constraints. Therefore, supplementary multi-variable data derived by advanced modelling and remote sensing were considered to overcome limitations in the conceptualization of processes, as well as to handle potential biases in precipitation forcing. However, a more realistic representation of internal processes may have some negative effects on streamflow simulation. Different scenarios (1-4) were therefore analysed to explore the role of constraining data for model calibration and to characterize the benefits and trade-offs in performance.

Similarly to other studies (e.g., Mayr et al., 2013; Finger et al., 2015; Tarasova et al., 2016; Chen et al., 2017a; Zhang et al., 2025), the use of additional variables led to a reduction in streamflow performance metrics, which was largely offset by improvements in underlying process characterization. The latter were both in terms of accuracy in the representation of individual processes, with AET being underestimated when not constrained, and prevention of internal process compensation, with glacier melting otherwise appropriately switched off due to the underestimation of precipitation. It is noteworthy that there are few hydrological studies in cold environments that integrate AET into the calibration (van Tiel et al., 2020), although it is a significant component of the water budget for basins in the monsoon regime (e.g., Fugger et al., 2024).

Specifically, in Scenarios 1 and 2, although few lumped parameters were optimized, a satisfactory representation of magnitude and patterns of AET (both scenarios) and glacier storage change (Scenario 1) was obtained during calibration period, proving that the model structure was able to accommodate this additional information. While the reference AET was also captured in validation, for glacier water loss the simplified approaches did not reproduce the increasing rates retrieved from satellite data in other studies, and which have no correlation with temperature in the meteorological dataset (see Sect. 4.2.1). However, the impact of this error on streamflow at the basin scale remains marginal. This acceptable reproduction of AET and glacier loss was found to potentially hinder the simulation of significant streamflow characteristics, such as the timing of rising limb in the seasonal hydrograph. Specific modelling issues behind this shortcoming are commented in Sect. 3.2. Furthermore, since melting dynamics are influential on the seasonal hydrograph (e.g., Mackay et al., 2018), possible inadequacies in the conceptualization of the process, in the attribution of a priori parameters and in the spatialization of the forcings may have played a role. The effect of data inconsistency cannot be ruled out either, with calibration criteria directing the model to compensate for them in a competitive manner. However, using a two-parameter, time-invariant precipitation adjustment formula (Scenario 1C) was sufficient to adequately reproduce the seasonal hydrograph together with additional reference data. Finally, in Scenarios 3 and 4, the model simulated well the streamflow response, but having







omitted the contribution of glacier melt and significantly underestimating AET due to a poor precipitation adjustment. These latter scenarios are therefore not indicative for understanding the basin behaviour and for correctly quantifying the main terms of the water balance. It should be noted that these are the only scenarios in which the calibration returns a parameter – related to the processing of model forcings – that collapses to the boundaries of the range (Tab. S1).

Overall, despite the inherent data uncertainties and the simplified processes conceptualization, the model has shown potential to reproduce several key features of the streamflow dynamics. In this regard, the magnitude of the average annual flood peaks was well simulated, which is not trivial in the Himalayan basins (e.g., Singh et al., 2016b; Khadka et al., 2020; Wang et al., 2021; Saddique et al., 2022; Nazeer et al., 2022). Specifically, the efficiency index used for flow peaks does not drop below 0.94 between calibration and validation, even in the Scenario 4 where it was not integrated into the objective function. Ultimately, the assumptions underlying the hydrological modelling are supported to some extent by the overall satisfactory performances, while the calibration experiments indicate to properly evaluate the modelling configuration, also considering pros and cons of adding specific data and processes in relation to the application objectives.

#### 4.2.1 Glacier water loss simulation

The probable discrepancy between modelled and observed rate of glacier mass loss during validation period may be due to inaccuracies in climate forcings, oversimplification in melting conceptualization relying on temperature-based approach, as well as the control of non-climatic factors, such as glacier topographic and morphological attributes (e.g., Singh et al., 2016a; Barandun and Pohl, 2023). Focusing on modelling, not only the conceptualization itself, but also its specific implementation (e.g., the coarse spatial resolution) played a role, including a priori setting of sensitive parameters. For example, the threshold temperature  $T_0$  can prevent ablation regardless of the actual net shortwave radiation, which is the dominant driver for glacier melting and could be appropriately integrated in the model (Hock, 2003, 2005), while at high altitudes the air temperature tends to be less informative about the incoming energy inputs (e.g., Barandun and Pohl, 2023).

While on the one hand assessing glacier changes and their hydrological impacts would therefore require a more complex representation, on the other hand ice melt remains a minor component of streamflow in the study basin (Fig. 8-9). Here, two simplified and parsimonious conceptualizations were implemented to simulate glacier-sourced runoff. The first one considers static glacier areas, while in the second one an analytical implementation of the V-A scaling relationship was proposed, suitable for hydrological modelling and not computationally and data demanding. Similar to other studies (e.g., Naz et al., 2014; Duethmann et al., 2015; van Tiel et al., 2018; Tsuruta and Schnorbus, 2022), using a static or dynamic approach was found to have limited impact on reproducing the observed streamflow, due to the low ice-covered fraction and glacier-sourced runoff contribution in the study basin, in addition to the relatively short simulation period which limits the effect of glacier changes.







### 4.3 Local hydrological insights: comparison with other studies

The model performed satisfactorily especially under average flow conditions (reported via  $KGE_{sqr}$  index) and adequately represented the dominant underlying processes, with outcomes that can be considered plausible within modelling simplifications and assumptions made on data accuracy. In this perspective, this modelling study contributes to the understanding and prediction of basin response in the region of interest. Calibration against multiple independent data sources allowed to obtain reliable and significant hydrological insights, for example in terms of water partitioning and interseasonal redistribution of precipitation contributions to streamflow. It is noteworthy that the basin area corresponds to a scale ( $\sim 10^3$  km²) which is challenging to investigate in water balance studies, because this intermediate dimension is particularly exposed to observational gaps and significant uncertainties (Dorigo et al., 2021; Hoeltgebaum and Dias, 2023), as well as a 20-year modelling analysis on an hourly scale is not commonly performed, since studies using this temporal resolution typically cover shorter periods (van Tiel et al., 2020).

The hydrological results are consistent with those of other multi-year simulations available in literature and carried out on the largest Alaknanda River basin at the confluence with the Bhagirathi River (e.g., Singh et al., 2023b; Rautela et al., 2023a, 2023b; Kavya et al., 2025). The performances are not superior to those obtained here, despite the more complex modelling of some processes, as well as difficulties were encountered in capturing some peaks and recession limbs in the hydrograph. In Kavya et al. (2025), the fully distributed physically based WATFLOOD model was used to simulate the surface runoff data from the ERA5-Land reanalysis, as an alternative to the stream gauge observations. Since the reanalysis surface runoff is not inclusive of the glacier melt contribution, the latter was not reproduced. In Rautela et al. (2023a, 2023b), semidistributed conceptual models, namely SWAT and SRM, were employed to simulate processes at the scale of elevation bands, with precipitation and temperature data spatialized via lapse rates. While snow dynamics modelling was quite complex, the contribution of glacier melt was not considered. In Singh et al. (2023b), the fully distributed conceptual SPHY model was used to evaluate the snow and glacier melt runoff considering observed streamflow and remotely sensed snow cover and glacier area variations. Modelled snow and glacier melt have peaks in June-July and August-September, respectively, thus being quite consistent with the results obtained here (Fig. 9). An indication of the temporal distribution of the glacier-sourced streamflow is also provided in the observational study by Kumar et al. (2018b), who measured meltwater from a glacier in the Alaknanda River basin and identified the monthly runoff peak in August. Finally, in terms of water balance, the outcomes of our study (Fig. 8) are quite consistent with those of Rautela et al. (2023b), who quantified streamflow and AET at 62% and 35% of the total precipitation respectively, and the contribution of snow melt ranging from 20 to 24% of the total streamflow, with the latter being split between surface and subsurface runoff in terms of 1/3 and 2/3. The higher volumes of AET and the slightly lower contribution of snow melt compared to our study are attributable to the greater extension downstream in the basin they considered. In Singh et al. (2023b), although rainfall is still dominant in the streamflow generation, a very significant contribution is attributed to glacier melt (22%), which in their model



conceptualization includes meltwater from permanent snow- or ice-covered surfaces, resulting even higher than snow melt (14%).

The comparison with independent studies therefore supports the results obtained here in terms of characterization of water balance components and key hydrological processes. Similarities in performance with models having more complex conceptualizations for snow and glacier processes suggest that the simplifications adopted in this study had a limited effect on the hydrological simulations. In this regard, the proposed modelling approach can be a useful tool for exploring the hydrological behaviour of basins under similar conditions, providing particularly valuable insights in contexts of observational and knowledge gaps.

#### 4.4 Current limitations

In the current application of the model, some limitations must be highlighted, which are largely related to the scarcity and high uncertainty of the data and could be addressed in future studies. Specifically:

- the spatial downscaling of hourly meteorological forcings was not addressed and the bias correction was performed only for precipitation data through hydrological modelling at basin scale;
- for each variable a single dataset was considered, attributing the errors only to precipitation and neglecting other inherent data uncertainties;
- a basic approach was used for parameter calibration, based on a single-objective function embedding multiple criteria with a priori set weights;
- no specific solutions were implemented to remove excessive snow accumulation at high altitudes due to a locally poor positive degree-day sum.

Regarding the first point, coarse meteorological data are typically processed to obtain higher resolution estimates consistent 650 with local ground observations. The latter may therefore be informative for gauge-based bias correction, as well as for downscaling in the typical form of gauge-interpolated vertical gradients of precipitation and air temperature, recommended due the strong impact of elevation (e.g., Immerzeel et al., 2014; van Tiel et al., 2020) but whose effectiveness in accurately reconstructing meteorological fields at the large basin scale may be limited (e.g., Chen et al., 2017b; Yang et al., 2025). Ground observations in the mountainous areas may be unavailable or lacking in accuracy and representativeness (e.g., Wortmann et al., 2018; Mishra et al., 2021; Barandun and Pohl, 2023) and do not systematically constitute an added value compared to the meteorological model estimates (Lundquist et al., 2019). Gauge-based lapse rates are often poorly captured due to unavailability of elevational transects and, where stations are in valley areas, are essentially extrapolated, with significant impacts on simulations (e.g., Magnusson et al., 2011; Hegdahl et al., 2016; Wang et al., 2024). Alternatively, a common practice in hydrological models is to calibrate or fix lumped linear gradients (e.g., Finger et al., 2015; Van Beusekom and Viger, 2016; Wang et al., 2021; Ruelland, 2024), even though they might not reflect the actual distribution of 660 the considered variables (e.g., Ragettli et al., 2013; Tarasova et al., 2016). In the Himalayas, representing local elevation dependence of precipitation is particularly challenging, due to different types of orographic controls operating at different








scales and resulting in complex precipitation patterns (e.g., Barros et al., 2004). Observational and modelling studies (e.g., Singh and Kumar, 1997; Arora et al., 2006; Shrestha et al., 2012, 2019; Immerzeel et al., 2012, 2014; Baral et al., 2014; Dahri et al., 2016; Meher et al., 2018; Mimeau et al., 2019a, 2019b; Yadav et al., 2020, 2024; Dimri et al., 2021; Jiang et al., 2022; Regmi and Bookhagen, 2022; Wolvin et al., 2024) highlighted the inadequacy of the assumptions of monotonic increasing trend and linear variation, as well as significant changes in the relationship with season, precipitation type or intensity, aspect, elevation range, and location in different mountain ranges within the basin. At the storm scale, significant events such as cloudbursts occur mainly at altitudes of 1000-2000 m and therefore in specific and not particularly elevated areas in the Indian Himalayas.

In this study, weather station measurements were not considered, and ERA5-Land data with the original resolution were used, although they are not informative on the highly variable and small-scale dynamics that would be useful to capture for hydrological modelling of the study basin. This involved simulating the snowpack and glaciers dynamics at a very coarse scale compared to that at which the underlying processes operate. Given the rainfall-driven runoff generation under high flow conditions, it was hypothesized that averaging the small-scale variability of non-dominant melting processes would not significantly affect the overall large-scale representation of the basin response. Furthermore, as stated above, widely used downscaling methods, based on stationarity in the relationship between meteorological patterns and some predictors such as elevation, would likely be inaccurate at hourly resolution. This is also due to the lack of data and understanding of the underlying dynamics (e.g., difficulty in estimating a complex but reliable relationship between precipitation and altitude) (Johnson and Rupper, 2020), as well as to the inheritance of the coarse scale errors (Fowler et al., 2007). Similarly, and more significantly, a locally effective gauge-based bias correction could then plausibly be outweighed by the precipitation adjustment made to address the water imbalance at basin scale.

In any case, the coarse modelling resolution did not prevent from obtaining reliable simulations and capturing the relevant processes in the study basin, although this may also be due to the model's ability to compensate for inaccurate forcing fields (e.g., Magnusson et al., 2011). The model performances are expected to improve with the availability of good quality, higher resolution distributed datasets and an adequate gauge network providing accurate observations (e.g., Bannister et al., 2019; Evin et al., 2024), which can more effectively support the representation of processes at proper scales. In this perception, the proposed modelling procedure may benefit from the outputs of well-configured higher resolution meteorological models (i.e., kilometre-scale), adopting convection-permitting approaches (Lundquist et al., 2019), which rely on a finer representation of complex landforms. They have been found to improve the simulation of diurnal cycle of precipitation (e.g., Ahrens et al., 2020), localized phenomena (e.g., Collier and Immerzeel, 2015; Karki et al., 2017), and extreme rainfall events (e.g., Chevuturi et al., 2015; Karki et al., 2018) in the Himalayan regions.

Then, limitations were highlighted concerning the joint use of multiple data sources. Although it is reasonable to assume that the main source of error comes from precipitation, uncertainties in other data should be considered, also evaluated with the analysis of multiple datasets. More accurate data with characterized uncertainty can then be used profitably within an appropriate multi-objective calibration framework (e.g., Efstratiadis and Koutsoyiannis, 2010). The desirable extension of




considered datasets may also include other additional variables, primarily observations of snow dynamics, which can thus support a modelling of more adequate complexity.

Information on observed snow patterns can not only provide a more realistic characterization of melting rates, thus improving streamflow representation, but can also highlight possible model shortcomings. In this regard, occurrences of excessive snow accumulation here were reduced by the DDF enhancement with elevation, thus affecting only limited areas in 3 northern sub-basins. At the same locations, Bandyopadhyay et al. (2019) detected an increase in accumulation area ratio, although the elevation change in the ablation zone was such as to lead to an overall negative mass balance at the basin scale. While local occurrences could therefore be plausible, disregarding snow accumulation effects can lead to an incorrect representation of hydrological fluxes and water balance (for example, here precipitation adjustments were also influenced by snow dynamics). For these issues, conceptual hydrological models typically integrate 'ad hoc fixes' approaches (Freudiger et al., 2017; van Tiel et al., 2020), such as artificially enhanced snow melting (e.g., Burek et al., 2020) or simplified snow redistribution methods (e.g., Tarasova et al., 2016).

#### **5 Conclusions**

This study implemented a conceptual, semi-distributed hydrological model (MISDc-2L) to simulate the hydrological response in the water years 2001-2020 in the monsoon-dominated, glacier-influenced Alaknanda River basin, a major tributary of the Ganges. Specifically, a tailored and parsimonious conceptualization enhanced by using additional reference data was tested for a reliable (as well as feasible and efficient) flood modelling. Multiple scenarios were explored, differing in the data used to constrain model calibration, the methods applied to correct systematic precipitation errors, and the treatment of glacier melt – whether explicitly modelled or not. Despite significant input data uncertainties – particularly in precipitation – the model successfully reproduced key hydrological processes when constrained with multi-variable data, namely glacier stored water loss and AET.

The analysis showed that:

- Despite its simplified and parsimonious conceptualization, the model proved capable of reproducing observed streamflow during both the calibration and validation periods with a KGE of 0.88 and 0.83, respectively, for the baseline scenario. These increased to 0.93 and 0.92 when using a two-parameter precipitation adjustment formula. The model demonstrated greater reliability under average flow conditions and effectively captured significant features of the high flow regimes. However, its performance in simulating specific flash flood events was limited, primarily due to localized inaccuracies in the rainfall data.
- The model accurately reproduced reference estimates of glacier water loss and AET when additional data were embedded in the calibration framework. Using multiple reference datasets enhanced the model's ability to represent the internal behaviour of the hydrological system; however, it also introduced trade-offs in seasonal hydrograph estimation in some scenarios where bias in precipitation data was not sufficiently addressed.





- Parsimonious precipitation adjustments can significantly improve streamflow simulation, by handling biases in the original dataset. Model simulations enabled the quantification of precipitation underestimation for hydrological applications, with the extent of such underestimation varying by season and precipitation intensity.
- Glacier melt contributes marginally to overall streamflow, but its inclusion improves internal model consistency. Simple conceptualizations, such as temperature-driven melting combined with static or V-A scaling approaches for glacier evolution, may be appropriate for hydrological simulations in monsoon-dominated basins. Here, reference data available for calibration were well reproduced both in terms of magnitude and spatial patterns, but during the validation period the model did not capture the expected increase in glacier stored water loss.

The 20-year modelling analysis yielded hydrologically consistent estimates of the main water fluxes, refined through the integration of independent multi-variable data. This analysis enabled a deeper exploration of various aspects of the water cycle within the study basin – capturing seasonal and spatial patterns as well as interannual variability – and contributed to advancing process understanding in hydrologically heterogeneous, monsoon-dominated basins of the Indian Himalayas.

Overall, the study illustrates a practical modelling strategy for data-scarce Himalayan basins with similarly complex hydrological processes, offering valuable insights for regional flood forecasting and water balance assessment. Nonetheless, limitations persist due to coarse input data and simplifications in model structure, parameterization and calibration scheme, with non-dominant snow and glacier processes being highly generalized in this large-scale, flood-oriented application. Future work should focus on improving the spatial resolution of process simulation, enhancing the modelling for adequacy and parsimony (by adding justified complexities to the conceptualization and improving the parameterization also through sensitivity analysis), and applying multi-objective calibration techniques using uncertainty-characterized datasets – including reference information on snow dynamics.

The availability of accurate high-resolution precipitation data remains essential to improve the predictability of high flows

during the ISM season, regardless of the complexity of the hydrological model. The inability to capture localized heavy
rainfall events constrains the use of hydrological models as predictive tools for flood forecasting and hinders progress in
understanding the hydrological response to extreme precipitation in the Indian Himalayas.

# Appendix A

Volume-area (V-A) scaling is a widely used approach for estimating the total ice volume of large sets of glaciers and its temporal changes (Bahr et al., 2015). The method is based on the following relation:

$$V = cA^{\gamma} \,, \tag{A.1}$$

where V and A are the volume and surface area of glaciers (which can be expressed in km<sup>3</sup> and km<sup>2</sup>, respectively),  $\gamma$  is the dimensionless scaling exponent, and c is the multiplicative scaling coefficient [km<sup>3-2 $\gamma$ </sup>]. According to Bahr et al. (2015), the scaling exponent can be fixed to a theoretical constant ( $\gamma = 1.375$ ), while the scaling coefficient is a variable. More generally,  $\gamma$  and c can be evaluated with different approaches and at different spatial scale, which has led to a wide range of estimates


(e.g., Radić and Hock, 2010; Huss and Farinotti, 2012; Grinsted, 2013). A global mean value of the scaling coefficient  $c = 0.034 \text{ km}^{3-2\gamma}$  was proposed by Bahr (1997). The impact of the error in c is reduced when the method is applied to large sets of glaciers (Bahr et al., 2015). According to Bahr (2011), a sample size of ~100 glaciers appears sufficient and, in many applications, far fewer glaciers may be reasonable.

The V-A scaling is often used for efficient representation of glacier dynamics within hydrological models (e.g., Lutz et al., 2013; van Tiel et al., 2020; Yang et al., 2025), as it can be applied at the scale of large basins, requires minimal input from readily available data, and generalises changes in glacier extent without modelling individual glaciers. Typical use in hydrological models is based on the estimation of the volume variation on a certain time interval (e.g., Luo et al., 2013; Valentin et al., 2018; Pesci et al., 2023), from the area at the beginning of the period and the subsequent ice melting or accumulation (in practice the latter is applied to the previous average ice thickness). Glaciers area corresponding to the updated volume is then computed according to Eq. (A.1).

Here, an analytical formulation was considered to continuously simulate the glaciers area evolution as a function of ice melting, based on a simplified representation suitable for hydrological applications, thus reducing the approximations of the approach just illustrated. While such approximations generally did not have a significant impact on the generated runoff, the proposed implementation does not imply increased computational demands. In the following, reference is made only to the case of glacier melting, due to the prevailing dynamics observed in the study basin and to the interest in streamflow modelling; however, this can easily be generalized to include also occurrences of glacier accumulation. Parameters  $\gamma$  and c are assumed to be known for the glacier population in the study basin. Specifically, the global values  $\gamma = 1.375$  and c = 0.034 km<sup>3-2 $\gamma$ </sup> were used in this application, even if slightly different from those estimated in literature in the study region (e.g., Sattar et al., 2019). The glaciers area evolution is assessed separately for each of the sub-basins, which should be outlined in such a way as to include an adequate number of glaciers. Ice melting is here calculated with the degree-day method for each grid point of meteorological dataset, considering only those to which glaciers belong. Meltwater is then averaged to obtain a lumped value at the sub-basin scale to be associated with the ensemble of glaciers which occupy the ice-covered fraction. Melting causes a change in thickness and modifies the aggregate volume of the considered glaciers.

The infinitesimal variation in thickness at time  $\tau$ ,  $dh(\tau)$ , is here made to correspond to the variation in volume:

$$dV(\tau) = dh(\tau) \cdot A(\tau) , \tag{A.2}$$

from which it can be obtained in the interval 0-t:

$$h_t - h_0 = \frac{c\gamma}{\gamma - 1} \left( A_t^{\gamma - 1} - A_0^{\gamma - 1} \right). \tag{A.3}$$

The glacier area evolution can be related to the melting process considering:

$$M_{\text{ice},t} = -\frac{\rho_i}{\rho_{\text{tr}}} (h_t - h_0) \cdot 10^6$$
, (A.4)

since a negative variation in h [km] can be made to correspond to the cumulative ice melting  $M_{\text{ice},t}$  [mm], while  $\rho_i$  and  $\rho_w$  are the density of ice and liquid water, respectively. This results in:

$$A_t = \left[ A_0^{\gamma - 1} - \frac{\gamma - 1}{c\gamma} \cdot \frac{\rho_W}{\rho_i} \cdot M_{\text{ice},t} \cdot 10^{-6} \right]^{\frac{1}{\gamma - 1}}.$$
 (A.5)

Equation (A.5), which is applied here at the sub-basin scale, can hold at any spatial scale for which Eq. (A.1) is assumed to be valid. For hydrological applications, it may be of interest to simulate the evolution of the glacierized fraction of the sub-basin,  $W_{g,t}$ , which is used to weight the current glacier-sourced melting flux:

$$W_{g,t} = \left[ W_{g,0}^{\gamma - 1} - A_{\text{sub}}^{1 - \gamma} \cdot \frac{\gamma - 1}{c\gamma} \cdot \frac{\rho_w}{\rho_i} \cdot M_{\text{ice},t} \cdot 10^{-6} \right]^{\frac{1}{\gamma - 1}}, \tag{A.6}$$

where  $W_{g,0}$  is the initial glacier-covered fraction and  $A_{sub}$  is the total sub-basin area.

# Data availability

NASADEM elevation (https://lpdaac.usgs.gov/products/nasadem\_hgtv001/), ERA5-Land reanalysis (https://cds.climate.copernicus.eu/datasets/reanalysis-era5-land?tab=overview), RGI glacier outlines (https://nsidc.org/data/nsidc-0770/versions/7), and GLEAM evaporation (https://www.gleam.eu/) data are publicly available. Glacier stored water loss data are reported in the open access research paper by Bandyopadhyay et al. (2019). Streamflow data of Alaknanda River at Rudraprayag gauge were provided by CWC at our request and are classified.

#### 805 Author contribution

Conceptualization: CM, SB, DDS, SS, AS, AA, VM. Formal analysis, investigation, methodology, software, visualization, writing (original draft preparation): DDS. Funding acquisition: SB, SS. Project administration: SB, SS, AS, AA, CM. Resources: SS, AS, AA. Supervision: CM, SB. Writing (review and editing): SB, VM, SS, FB, AS, AA, SG, FA, CM.

## **Competing interests**

Some authors are members of the editorial board of Natural Hazards and Earth System Sciences.

#### Acknowledgements

This study was carried out in the framework of the FLOSET Project "Probabilistic floods and sediment transport forecasting in the Himalayas during the extreme events", funded in the context of the 'ITALY-INDIA JOINT SCIENCE AND TECHNOLOGY COOPERATION CALL FOR JOINT PROJECT PROPOSALS FOR THE YEARS 2021 2023'.

The authors sincerely thank the Central Water Commission - CWC (New Delhi, India) for providing the streamflow data that were used in this study.

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
