# Peer review of "Harnessing multi-source hydro-meteorological data for flood modelling in a partially glacierized Himalayan basin"

_EGUsphere, 2025_

## Author Comment (AC1)

Hydrological modelling in complex-terrain areas is highly challenging due to uncertainties in model structure, parameters, and input data, further complicated by the issue of equifinality, which often obstructs accurate interpretation of hydrological processes. This study utilizes multi-source data to calibrate a semi-distributed hydrological model, thereby providing better constraints on key hydrological processes. The findings offer valuable insights and contribute to advancing the understanding of water cycle in complex terrains. Main concerns are listed as follows.

We thank the reviewer for the constructive comments and suggestions, which gave us the opportunity to strengthen our manuscript. We believe that we have addressed all concerns and incorporated all suggestions as detailed in the point-by-point response below.

1) Regarding the hydrological and calibration strategy, more details are needed. Key information, such as the inputs and outputs of the model, model integration time step, temporal resolution of reference data used for model calibration, and the number of iterations for each calibration, needs to be clearly specified.

We will further specify the requested information in the revised manuscript where necessary, as illustrated below:

- inputs and outputs of the model: the hydrological model was widely used (L166-168) and, in order not to further lengthen the main text, it is described in detail in Section S1 of the Supplement (L180-182). However, we agree that it is appropriate to make the required information explicit by inserting the following sentence in Section 2.5 at L174: "It uses precipitation, air temperature and PET (here derived from $ET_0$) as forcing data, and reproduces various internal states and hydrological fluxes, including outgoing streamflow and AET";
- model integration time step: as stated in L174, the model was applied at hourly time-step;
- temporal resolution of reference data used for model calibration: the sentences in L285-288 will be slightly modified to better specify this information, as reported below. "KGE is the Kling-Gupta efficiency index (Gupta et al., 2009), computed between observed and simulated streamflow time series, while APFB is the annual peak flow bias proposed by Mizukami et al. (2019). $AET_{sim}$ and $AET_{ref}$ indicate the simulated and reference cumulative AET volume in the period at basin scale, respectively, while similarly $IMV_{j,sim}$ and $IMV_{j,ref}$ are the simulated and reference glacier-sourced meltwater volume for the $j$-th sub-basin aggregate". Slight changes will also be made to the material sections to better clarify the resolution of the available data. Specifically, streamflow data was collected once a day at a fixed known time (L116), while for glacier water loss, the mass change attributed to the period 2000-2014 was available (L158, see also in AC2 the updated version of the sentence in the response to comment on L156-165);
- the number of iterations for each calibration: technical details on the calibration algorithm configuration will be included in Section S1 of the Supplement, as reported below. "For model calibration, a single-objective global optimization algorithm was applied, specifically the Covariance Matrix Adaptation Evolution Strategy, CMA-ES (Hansen et al., 2003). Several CMA-ES control settings were not altered from their defaults, except for three termination criteria. Specifically, "TolFun" and "TolX", which are mainly associated with the objective function values and the standard deviation of the normal distribution used to sample the parameter values, respectively, were set at $10^{-4}$. Furthermore, the maximal number of function evaluations ("MaxFunEvals") was fixed at $10^5$. The model parameters were encoded in the range [0; 10] so that they would presumably have similar sensitivity. For this purpose, a linear or logarithmic transformation was used, depending on the original parameter range. For each parameter, the initial solution point was generally chosen equal to the intermediate value (i.e., 5), and the initial

standard deviation for the sampling distributions was set to one-third of the parameter range (i.e., ≈ 3.33)".

2) In terms of validation, the comparison of water balance components across different scenarios should be strengthened, particularly with respect to AET and glacier meltwater loss.

We will conduct a more systematic analysis of the water balance in the different simulation scenarios in Section 3.3, also summarizing the points of interest already introduced in Section 3.1 ("Calibration and validation analysis"). See also the response to comment 15, where an indicative figure for comparison is provided. With reference to the AET and glacier melting components, it is worth noting that the differences between scenarios are mainly explained by the inclusion or not of these variables in the calibration, rather than from specific model configurations. This results in some scenarios being very similar to each other in terms of water volumes, which is why some analyses were illustrated with reference only to the baseline, having previously highlighted how specific scenarios differed from it.

3) Given that runoff observations are rare in complex terrain areas, while satellite-observed evapotranspiration and glacier changes are more readily available, it is recommended to consider another scenario that explores the model performance when calibrated without using observed runoff (i.e., utilizing only AET and glacier meltwater loss), which may have broader applicability in complex-terrain areas.

In the revised version we will integrate the scenario recommended by the reviewer (Scenario 5). Here we anticipate that the simulation has returned some interesting results, with the AET and glacier meltwater loss data proving effective in correcting for precipitation and thus representing a water balance which does not deviate excessively from the other scenarios, despite significant limitations in simulating streamflow without specific constraints. The table below shows the updated results in terms of efficiency indices, integrating Scenario 5.

| Period | $\varphi$ | Scenario | | | | | | | |
|---|---|---|---|---|---|---|---|---|---|
| | | 1 | 2 | 3 | 4 | 5 | 1B | 1C | 1D |
| CAL | KGE | **0.882** | **0.894** | **0.907** | **0.911** | 0.656 | **0.904** | **0.930** | **0.878** |
| | Eff$_{APFB}$ | **1.000** | **1.000** | **1.000** | 0.940 | 0.131 | **1.000** | 0.999 | **1.000** |
| | Eff$_{AET}$ | **1.000** | **1.000** | 0.651 | 0.642 | **1.000** | **1.000** | 0.999 | **1.000** |
| | Eff$_{IMV}$ | **0.993** | 0 | 0 | 0 | **0.994** | **0.995** | **0.982** | **0.993** |
| VAL | KGE | 0.828 | 0.823 | 0.856 | 0.860 | 0.582 | 0.873 | 0.918 | 0.831 |
| | Eff$_{APFB}$ | 0.966 | 0.955 | 0.982 | 0.935 | 0.301 | 0.952 | 0.951 | 0.968 |
| | Eff$_{AET}$ | 0.998 | 0.998 | 0.648 | 0.640 | 0.995 | 0.999 | 1.000 | 0.998 |

**Table 3. Efficiency indices considered for the objective function, during the calibration (CAL) and validation (VAL) periods, for the different scenarios. In bold the values of the metrics optimized during the calibration.**

Some minor points:

4) The title focuses on flood modelling, which is inconsistent with the paper's limited coverage of this topic. It is suggested that the title be revised to more accurately reflect the actual content of the paper.

We agree with the reviewer on the lack of in-depth investigation at the scale of flood events, due to limitations in the representation of extreme and localized rainfall in the coarse meteorological dataset. Hydrological modelling is in any case oriented towards the high flow regime, with satisfactory streamflow performances offering insights into flood response dynamics and indicative of the potential as a tool for flood simulation. Accepting the suggestion, we will change the title to the following:

"Harnessing multi-source hydro-meteorological data for high flows modelling in a partially glacierized Himalayan basin".

5) Line 15, "the model ... precipitation input", observed streamflow was also used for model calibration.

We will modify the sentence as follows: "The model was calibrated using multi-variable data, including satellite-based glacier water loss and actual evapotranspiration in addition to streamflow, also to address bias in the precipitation input".

6) Line 18, "Multi-variable calibration improved...", Multi-variable calibration not always improved the simulation of hydrological fluxes, as evidenced by the poorer streamflow simulation in Scenario 1 compared to Scenario 4. However, multi-source data calibration can provide a more plausible representation of hydrological processes.

We will modify the sentence as follows: "Multi-variable calibration provided a more plausible representation of hydrological processes and highlighted the value of using complementary satellite-based information in data-poor mountain regions".

7) Line 184, the method for rainfall-snowfall partitioning should be provided.

The partition method is applied in a basic version, with precipitation classified as snow if the air temperature is equal or below the threshold, and as rain otherwise. We will go into more detail by modifying the part of interest as follows: "A common threshold value was applied to air temperature both to classify simultaneous precipitation as rain or snow and to reproduce melting processes within the well-known degree-day method, with the snowpack acting as temporary water storage. Specifically, melting over snow- and ice-covered areas was simulated based only on hourly air temperatures as: ...".

8) Does DDF in Equation 3 represent the degree-day factor for snow, ice, or both?

As stated in L204, Equation 3 is referred to snow, but due to the linearity in Equation 2 a relationship of the same type describes the variation of $DDF_{ice}$ with altitude (with only the rescaling of the minimum and maximum DDF values).

9) Line 227-229, it would be appreciated to provide a figure or table related to these results.

We will add a figure in the Supplement (Fig. S4), which we also provide here.

[Figure]

**Figure S4. Annual distribution of ERA5-Land precipitation and gauge streamflow volume at basin scale (on the top) and corresponding values of the runoff-to-precipitation ratio (on the bottom).**

10) Why are glacier mass losses not simulated in Scenarios 2-4?

As stated in L290-293, glacier mass losses are not simulated in the absence of specific constraints (i.e., $\Phi_4$ in the objective function), mainly due to underestimation of precipitation which would tend to be compensated by overestimation of modelled ice melt, considering also that the process that is thus neglected has a limited impact on the water budget at basin scale.

11) Line 289, the basis for determining these weights requires clarification.

We will restructure the sentence in L259-263 by providing additional details to clarify the basis for determining the weight values: "Despite some disadvantages (Efstratiadis & Koutsoyiannis, 2010), such an embedded multi-criteria calibration approach is widely used (e.g., Gao et al., 2017; van Tiel et al., 2018; Mei et al., 2023), adopting suitable weights for an acceptable trade-off between the different objectives based on the model purpose. The weights allow the user to give different priorities to certain objectives and, with respect to these, they should reflect factors such as relative importance, reference data reliability, and actual differences in scale. Here, the practice of seeking a balanced solution by performing prior tests with variable weights was followed (e.g., Viviroli et al., 2009; Tarasova et al., 2016; Sleziak et al., 2020; Ruelland, 2024; Wagner et al., 2025), considering how the individual components were simulated".

12) Line 423-424, it is suggested to provide the root mean square errors for calibration and validation periods, respectively.

We will provide the metric values as requested. It should be noted that in the updated version this assessment will be done with reference to version 4 of GLEAM. The latter will be considered due to the comments subsequently provided by the second referee. The sentence will be changed as follows: "Figure 4 presents monthly timeseries of basin-scale averaged AET, provided by GLEAM v4 and the hydrological model for Scenario 1. A good agreement was obtained, with a root mean square error of about 3.1 and 2.8 mm month$^{-1}$ during calibration and validation periods, respectively".

13) Figure 5, it is recommended that the mean annual AET for both the calibration and validation periods be presented separately.

We thank the reviewer for pointing this out. AET (both simulated and reference) was found to be rather stationary, with no significant differences in the calibration and validation sub-periods (see also Fig. 4). This also applies to spatial patterns, as highlighted by the figure below, which aligned with the reviewer's request but was then replaced with Fig. 5 present in the manuscript and referred to the entire study period. Agreeing that it is appropriate to provide this information, we will modify the related sentence to: "Figures 5a and 5b illustrate spatial distributions of mean annual AET at sub-basin and pixel scales for MISDc-2L and GLEAM v4 data, respectively, over the entire study period since there are no significant differences between the calibration and validation years". The final version of Fig. 5, which now integrates three panels and returns AET differences at the sub-basin scale, is provided in AC2. Even using GLEAM v4, no significant differences in spatial patterns were observed between the two periods.

[Figure]

14) Figure 6, it is recommended to incorporate reference values for glacier mass loss into this figure.

We will insert an updated composite figure (see below) with a second subplot representing the cumulative distribution of simulated glacier water losses, where the reference data for comparison are also reported.

[Figure]

**Figure 6: Simulated glacier water loss in the study period, for different sub-basin aggregates according to Scenario 1: annually distributed (a) and cumulative (b) values, the latter compared with the geodetic reference for the calibration period.**

15) It is recommended to compare the mean annual water balance across different scenarios, following the format of Figure 8.

We will integrate the suggested analysis, also through the indicated figure, which we report below.

"Compared to the baseline, the other simulation scenarios show deviations in terms of mean annual water balance (Fig. 9) which can be explained by various elements previously highlighted. The precipitation volume is mainly governed by which terms ($I_{melt}$, AET, and $Q$) are simulated and considered in the multi-variable calibration. The distribution of total precipitation between rainfall and snowfall is obviously analogous in the scenarios that adopt the adjustment described by Eq. (4), while Scenarios 1B and 1C are distinguished by a higher fraction of solid precipitation. Scenario 1C, which has a more flexible precipitation adjustment, best captures the observed streamflow volumes (see the term $\mu_{sim}/\mu_{obs}$ close to 1 both in calibration and validation) resulting in a lower precipitation increase among all the baseline variants. The consistency across scenarios of snow and glacier modules parameters translates in very similar ice melting and snowpack accumulation volumes, depending on whether the contribution of the glaciers is considered or not. Similarly, the AET component remains at approximately the same values depending on whether it was included in the objective function or not. The configurations of the individual scenarios have a significant impact on the distribution between surface and subsurface runoff, with the latter being reliably represented as the major contributor to the total streamflow (except in Scenario 5 where the overestimation of infiltration excess was previously highlighted)".

[Figure]

**Figure 9: Simulated mean annual water balance for the study basin, according to Eq. (11), for the different scenarios.**

---

## Author Comment (AC2)

In this work, the authors present an extended version of a semi-distributed hydrological model and evaluate its performance across multiple processes in the complex, monsoon-dominated regions of the Indian Himalayas. The manuscript is generally well written, and the results are discussed in depth. However, I have few methodological concerns that should be clarified prior to publication.

We are grateful to the reviewer for the insightful and constructive feedback on our paper, which have significantly helped us to enhance the quality of the manuscript. We greatly appreciate the valuable contribution in identifying critical areas for improvement. However, it was possible to address some concerns only to a limited extent, due to data scarcity issues that are a peculiarity of the study area. Streamflow measurements are typically lacking in mountainous and transboundary regions, with limitations related to the practical difficulties of monitoring in such environments, as well as public access to measured data. Therefore, the availability of local information that would be essential for hydrological modelling (including, for example, data from internal stream gauges or about anthropogenic influences) is far from a given. Knowledge gaps make streamflow reproduction challenging, which in our opinion gives value and usefulness to the contribution of the methodological choices and the satisfactory local results of the study. We also point out that the model was calibrated with a stochastic optimization algorithm rather than Bayesian inference. This resulted in having a single set of parameters for each of the 8 current scenarios: to derive a rigorous quantification of parametric uncertainty from our elaborations, the use of a Bayesian approach would still be essential. The basic calibration procedure, as well as some simplifications due to data scarcity, have also been discussed in the section dedicated to the known limitations of the study. As further detailed below, we have carefully addressed the reviewer's comments to the extent reasonably possible, aiming to seize the full opportunity to improve the clarity and accuracy of the manuscript.

Novelty of the study: The authors should more clearly articulate the key scientific advances of this work. Extending and evaluating an existing hydrological model and exploring multiple calibration approaches are important objectives, but the manuscript does not sufficiently clarify what constitutes the main innovation compared with existing literature. This should be clearly stated, ideally in the introduction, so that the contribution of the paper is evident from the outset.

We thank the reviewer for pointing this out, allowing us to more clearly explain in the introduction the advances and contributions provided by our study. The latter implements and tests a practical modelling strategy which combines several methodological solutions, some of which are original, aimed at addressing specific challenges due to data scarcity in complex terrain and poorly monitored regions. In general, the study illustrates the implementation of various methods, including new ones, analyses the outcomes of their application, and provides contributions on still open topics such as the proper use of additional hydrological variables for model evaluation and calibration (e.g., Wagner et al., 2025). Furthermore, in the study context, the results obtained at the local scale also offer valuable insights into understanding the hydrological behaviour and streamflow generation processes.

These contributions in terms of methods and results will be better highlighted by inserting the following in correspondence with L89 and replacing the final part of the introduction:

"More generally, the integration of multi-variable datasets within hydrological modelling still presents several challenges and open questions, with a variety of calibration and validation strategies that have attempted to best exploit this potential (e.g., van Tiel et al., 2020; Wagner et al., 2025).

In this perception, the objective of this study was to develop and evaluate a parsimonious, semi-distributed hydrological modelling approach for simulating streamflow under high flow regime in the partially glacierized, monsoon-dominated basins of the Indian Himalayas. The modelling approach was tested in the Alaknanda River basin in northern India, where the ISM causes frequent flash floods, sometimes with disastrous effects (e.g., Joshi and Kumar, 2006; Rautela et al., 2023a, 2023b). The

feasibility of integrating reanalysis and satellite data was investigated, given the scarcity of ground-based information. Specifically, the proposed modelling approach was based on the implementation of 1) a specifically tailored hydrological model, and 2) a multi-variable and multi-response model calibration.

Regarding the first point, a conceptual model well-suited for rainfall-induced floods was modified with a tailor-made snow module and the addition of a static and a dynamic glacier module. The snow and glacier modules are characterized by low complexity and minimal data requirements, as well as by the proposal of original methods regarding the variability of melting parameters with altitude and the analytical formulation of melting-induced glacier evolution based on volume-area scaling. The model thus adapted was applied here for the first time in glacier-influenced basins and tested as a tool for reproducing streamflow dynamics in the study region. Regarding the second point, the modelling framework was enhanced by using multiple reference data in calibration, including satellite-based glacier water loss and actual evapotranspiration, to address bias in precipitation input and improve water balance representation. It is noteworthy that there are few hydrological studies in cold environments that integrate actual evapotranspiration into the calibration (van Tiel et al., 2020), although this is a significant component of the water budget for basins in the monsoon regime (e.g., Fugger et al., 2024). A specific objective function was proposed here by combining multiple criteria, which consider additional data characteristics and modelling objectives. The benefits for process representation and trade-offs with streamflow performance were evaluated through several simulation scenarios representing different calibration setups in terms of target variables. Other simulation scenarios were then devoted to comparing the impact of using the static or dynamic glacier module and the effects of different adjustment methods to handle biases in reanalysis precipitation.

The study therefore provides advancements in terms of specific methods and general results for the broader topics of fit-for-purpose hydrological modelling and integration of additional variables into calibration. The proposed modelling approach has proven to be valuable in a challenging context with complex dynamics and significant observational and knowledge gaps. Therefore, local results from this study contribute to understanding the streamflow response dynamics and water balance quantification in data-poor, monsoon-dominated Himalayan basins with glacierized headwaters. Although this study is somewhat preliminary due to the scarcity and high uncertainty of the data, the findings offer practical insights into the challenges of modelling hydrological fluxes and predicting floods in a region with increasing hydroclimatic risks".

ADDITIONAL REFERENCE:

Wagner, P. D., Duethmann, D., Kiesel, J., Pool, S., Hrachowitz, M., Ceola, S., Herzog, A., Houska, T., Loritz, R., Spieler, D., Staudinger, M., Tarasova, L., Thober, S., Fohrer, N., Tetzlaff, D., Wagener, T., & Guse, B. (2025). The unexploited treasures of hydrological observations beyond streamflow for catchment modeling. Wiley Interdisciplinary Reviews Water, 12(2). https://doi.org/10.1002/wat2.70018

Model calibration procedures and uncertainty discussion: Although calibration is presented as a central component of the study, there is no explicit investigation of model parameter uncertainty. It remains unclear how the different calibration methods and their respective constraints influence the uncertainty associated with parameter identification. Presenting only a single optimal value for each parameter, is restrictive, particularly given that calibration is one of the main goals of the paper. A more comprehensive analysis of parameter uncertainty would significantly strengthen the study.

We agree with the reviewer on the significant role of model calibration in our study and on the importance of uncertainty information in parameter estimates (along with that in the data and model structure, which are reasonably expected to contribute no less to the error budget in the study context). Here the calibration was performed with a stochastic optimization algorithm, rather than a Bayesian

inference algorithm designed to explicitly quantify parametric uncertainty via posterior probability distributions. Optimization algorithms are focused on locating a single global optimum of the objective function by estimating the corresponding single set of parameters. Despite this limitation, such an approach allowed to reduce the computational costs associated with the setup of the calibration experiments (search space dimensions, model time step and calibration window length, number of simulation scenarios). Specifically, the standard version of the Covariance Matrix Adaptation Evolution Strategy (CMA-ES) was used, which has been successfully tested in several hydrological applications. CMA-ES employs a multivariate Gaussian as the search distribution and updates its parameters, including a covariance matrix (C) that controls the shape, to generate successful search directions. A valid example of using standard CMA-ES computations to produce the posterior distribution of parameters (by the way, not necessarily Gaussian) is the study by Grayver and Kuvshinov (2016). In their work, the local information about the current search distribution provided by C at the optimum, further processed through a proper rescaling, contributes to initialize the proposal distribution and improve the efficiency of a Bayesian method, which is used in a subsequent step for simultaneous parameter estimation and rigorous uncertainty quantification. Such a hybrid strategy, combining CMA-ES optimization with Bayesian sampling to derive parametric uncertainty, is well beyond the scope of our study.

Here the calibration experiments compared different scenarios analysing their implications on streamflow performance and on the representation of hydrological processes. The calibration approach employed has proven useful for this investigation and more generally for the study context; however, we acknowledge its limitations, which are highlighted in Section 4.4, and do not recommend it as a general method. In this sense, it would be appropriate that for the scenario deemed ideal based on the purposes, a calibration can be carried out that goes beyond the search for a single optimal model realization, but which explicitly considers model structural, data and parameter uncertainty (a task that is far beyond the scope of this paper). In any case, we agree with the reviewer on the appropriateness of a comprehensive analysis of parameter uncertainty, as part of a less basic calibration approach (see L645). We have therefore made this point explicit in the discussion of the current limitations, integrating the sentence in L695 of the original manuscript as follows: "More accurate data with characterized uncertainty can then be used profitably within an appropriate multi-objective calibration framework for parameter estimation and uncertainty assessment (e.g., Efstratiadis and Koutsoyiannis, 2010)".

ADDITIONAL REFERENCE:

Grayver, A. V., and Kuvshinov, A. V. (2016). Exploring equivalence domain in nonlinear inverse problems using Covariance Matrix Adaption Evolution Strategy (CMAES) and random sampling. Geophysical Journal International, 205(2), 971–987. https://doi.org/10.1093/gji/ggw063

Model validation with observational data: The model validation against observed data appears limited. For example, the absence of independent river-flow validation at interior points of the basin is a limitation. Relying on a single outlet control point for such large catchments may mask the model's ability, or inability, to reproduce flow dynamics in locations within the basin unseen during calibration. Additional validation would help demonstrate the robustness and transferability of the calibrated model. Are the calibrated parameters spatially varying across the basin or constant in space? How this would influence the simulated results?

Regarding the first part of the comment, we agree that the lack of internal stream gauges (also representative of different hydrological behaviours within the basin) is limiting. However, in the study region the streamflow observations are extremely scarce, especially with increasing altitude and topographic complexity (in this perspective, another scenario will be added to represent the absence of hydrological information, at the suggestion of the first referee). It should also be noted that, in line with

the scope of the journal and modelling objectives, densely populated valley areas are of greatest concern in the Himalayan basins from a river flow-related risk perspective. In this sense, although additional upstream gauging stations are extremely valuable for model assessment, the information they provide may be of limited benefit in simulating streamflow at basin outlet (e.g., Lerat et al., 2012).

Regarding the second part of the comment, we highlight that for reasons of parsimony (and identifiability), the model parameters are lumped. However, we attempted to introduce original solutions in model parameterization to ensure spatial variability in process simulation, arriving at the presented configuration which led to satisfactory results. For example, a relationship was proposed to consider the dependence of degree-day-factors with elevation (see Section 2.5.1), as well as empirical relationships between river basin and streamflow response characteristics were used to parsimoniously represent the variability of convolution and propagation parameters across sub-basins (see Section S1 in the Supplement). This last point will be better highlighted in the main text in correspondence with L182.

ADDITIONAL REFERENCE:

Lerat, J., Andréassian, V., Perrin, C., Vaze, J., Perraud, J. M., Ribstein, P., & Loumagne, C. (2012). Do internal flow measurements improve the calibration of rainfall-runoff models? Water Resources Research, 48(2). https://doi.org/10.1029/2010wr010179

Artificial influences of the hydrological regime: Regarding river flow, the authors should also consider discussing whether artificial reservoirs, hydropower plants, or other human interventions influence the hydrological cycle in the study area, and, if so, how these anthropogenic factors are represented or accounted for in the model.

We thank the reviewer for the opportunity to clarify this point in the revised text. The model simulates only natural flows, similarly to many others typically employed in the study region. Its structure does not integrate components to reproduce anthropogenic effects and therefore would encounter limitations in basins where their impacts are significant with respect to the rainfall-runoff transformation. In this regard, it is appropriate to make some considerations both on the context and on the objectives of the study.

The presence of anthropogenic influences is clearly likely, in a region where water resources have significant potential for irrigation and hydropower. However, on the one hand their impacts are to be considered limited in the basin we have outlined, on the other hand, in view of a possible integration into modelling, the problem of availability and accessibility of the relevant data remains. In accordance with the cited study by Goteti and Famiglietti (2024), the area of our interest (Northern India, altitude above 1500 m, transboundary basins) is characterised, in addition to limited and "classified" streamflow data, by minimal groundwater extraction and water diversions, while both the density of dams and reservoirs and the corresponding storage capacity are low. Other hydrological studies on the Alaknanda River basin (closed further downstream), mentioned in Section 4.3 "Local hydrological insights: comparison with other studies" (Singh et al., 2023b; Rautela et al., 2023a, 2023b; Kavya et al., 2025), do not integrate the effects of operational hydroelectric power plants into the modelling, despite the main focus of their analyses being on water resources management rather than on high flows and flood response dynamics. In the study by Kavya et al. (2025), where it was even necessary to overcome the unavailability of streamflow data by replacing them with the ERA5-Land runoff simulations, they conducted a field trip to approach a hydropower agency that provided them with a year of streamflow observations for a gauged site. In Rautela et al. (2023b), the modelled streamflow was converted in hydroelectric potential (simulated HEP) and compared with data obtained for a power plant (plant-produced HEP) which is located downstream from our outlet. The authors highlight some limitations of

this validation approach, which does not consider, for example, operational and maintenance factors that play a role in determining the actual energy output.

In our study, hydrological modelling is oriented towards the high flow regime during the monsoon season, with satisfactory performance that provides insights into the streamflow response to extreme rainfall and is indicative of its potential as a tool for flood simulation. The study basin has faced repeated flash floods due to heavy rainfall events, including cloudbursts, and in such circumstances any possible anthropogenic effect has been found to be secondary. The analysis of the precipitation and streamflow time series does not show any alterations in the latter indicative of significative regulation effects (e.g., constant flow regime to variable precipitation inputs or sudden changes in the hydrographs that cannot be attributed to rain or snowmelt inputs), which would in any case impact the performance of a model that is designed for natural flows and not for highly regulated basins. Although not considering anthropogenic influences increases errors in model simulations, in a context such as the one under consideration it can be assumed that the contribution to the overall uncertainty budget is limited, compared for example to that attributable to meteorological inputs.

We will integrate this point into Section 4.4 "Current limitations", as reported below, although it can be assumed that our results and conclusions still apply. "". In L648 will be added "the model simulated only natural flows and therefore did not consider any possible anthropogenic influence on the hydrological cycle". In L709 will be added "Finally, the model structure did not integrate components to reproduce anthropogenic effects, nor were these considered in the analyses performed. While the streamflow in the study area is expected to be relatively less impacted by such factors (e.g., Goteti and Famiglietti, 2024), the latter are present but difficult to characterize due to the poor availability and accessibility of the relevant data. Here the modelling objective was to simulate the high flow regime during the monsoon season to gain insights into the flood response dynamics, in a basin subject to extreme events of such magnitude that the simultaneous effect of any anthropogenic influences can likely be considered secondary. The model achieved satisfactory performances in reproducing the streamflow time series, with the latter showing no evidence of significant alterations attributable to anthropogenic effects, especially regarding the generation and propagation of high flows. In any case, the model, as currently configured, is only indicated for mountainous basins with near-natural regimes".

**Specific comments:**

Line 118: How are these sub-basins defined? Are they delineated based on geomorphological attributes, land-use/land-cover, soil, geology, or another criterion?

Sub-basins were delineated based solely on elevation data, using GIS tools to calculate the flow direction matrix from the DEM (in this case, with the widely used D8 algorithm). From the resulting flow accumulation matrix, the river network was obtained by setting a threshold area for the creation of streams, and therefore the boundaries of the sub-basins. In the hope of improving clarity on this point, we will restructure the sentence in L117-118, adding this operator-defined information, to also facilitate reproducibility: "Digital elevation model data (NASADEM at 30-m resolution) were used to delineate the main drainage network, assuming a threshold area to form a stream of 200 km$^2$. The 19 sub-basins in Fig.1 were thus identified, having areas ranging between 160 and 778 km$^2$".

Line 137: It appears that S1 has not been introduced yet.

Yes, because the Supplement contains an initial section referring to the hydrological model that includes a first figure (indicated as S1 in the original version and not referenced in the main text). To resolve what appears to be an inconsistency, we will ensure that the figures referenced in the main text have indices that start from S1, making the necessary changes in the Supplement as well.

Line 145: Why is GLEAM v4 not used? It offers higher spatial resolution and extended temporal coverage.

In the calibration, the AET volume at basin scale (and therefore lumped) was used as a constraint, to correct precipitation in accordance with a realistic water balance. This target value did not differ significantly from GLEAM v3 to v4, nor was there a benefit from the extended temporal coverage as our study period was limited to 2020 by the availability of streamflow data. It is noteworthy that the spatial variability of the simulated AET is likely driven more by the meteorological forcings than by the lumped model parameters, limiting the benefits on the latter of a calibration metric that considers AET patterns. While we considered it not worth repeating the calibration experiments for the several scenarios, we agree that it may be appropriate to use the enhanced and later available GLEAM v4 for the assessment of the simulated AET patterns (especially the spatial ones, which have been previously evaluated only in qualitative terms). Therefore, we will update Section 2.3 as follows: "Actual evapotranspiration (AET) from the GLEAM dataset v3.8a at 0.25° spatial resolution (Miralles et al., 2011; Martens et al., 2017) was used as a basin-scale reference for model calibration and validation with a lumped approach. For further evaluation regarding spatial patterns, a later available version with an improved spatial resolution of 0.1°, namely GLEAM v4.2a (Miralles et al., 2025), was instead considered. Potential evapotranspiration (PET) was computed with a Priestley-Taylor (GLEAM v3) or Penman (GLEAM v4) equation and then converted into AET considering an evaporative stress factor. Specific parametrisations were implemented for ice- and snow-covered regions. GLEAM algorithm employs several forcing datasets, such as reanalysis radiation and air temperature, a combination of gauge-based, reanalysis and satellite-based precipitation, and satellite-based vegetation optical depth, as well as it assimilates satellite-based surface soil moisture". Similarly, we will update Section 3.3 with the new results (the figures of which we anticipate below) and related comments.

[Figure]

**Figure 4: Comparison between simulated (in red) and reference (in blue) monthly AET data at basin scale in the study period for Scenario 1.**

[Figure]

**Figure 5: Mean annual AET in the water years 2001-2020 for Scenario 1, according to semi-distributed hydrological model (a) and gridded reference (b). Panel (c) shows the differences at the sub-basin scale between the simulated and reference AET.**

Lines 156–165: Could you provide more details on the methodology? When you refer to "summary data," what exactly does this include? Are these spatially varying discharge time series for each sub-basin? At what temporal resolution?

To improve clarity on this point by adding appropriate details, we will modify the sentences in L157-164 as follows: "For this work, the study by Bandyopadhyay et al. (2019) was taken as a reference. Geodetic glacier mass balance data were calculated from elevation changes, evaluated on RGI outlines by the differences between two satellite-based high-resolution digital elevation models referring to the years 2000 (SRTM mission) and 2014 (TanDEM-X mission), respectively. The mass changes thus attributed to the period 2000-2014 were validated against previous studies on selected glaciers. Glacier mass balance values aggregated to the river basin scale were provided for the two main tributaries (Dhauliganga and Pindar) and the Upper Alaknanda (upstream of the confluence with Dhauliganga). These latter data, in terms of stored water loss, were used in this study as independent reference in the model calibration, for the corresponding three groups of the sub-basins outlined here. To be consistent with Bandyopadhyay et al. (2019), an ice density of 850 kg m$^{-3}$ was assumed, appropriate for converting geodetic glacier volume changes (Huss, 2013)".

In the original version, with 'summary data' we meant the values of the glacier mass variations at the scale of the river basin, with the latter corresponding to an aggregate of the sub-basins outlined here. There are no time series of data available, and the spatial scale is larger than that of the individual sub-basins.

Line 166: Could you clarify the elementary spatial unit used by the model for the different hydrological processes? Is it a 0.1° pixel?

Line 178: "Ice melting was simulated only…" On what basis are these grid points classified? Is this classification time-varying? Why is ice melting simulated only for these pixels and not for all pixels where snow is present?

We have merged these two comments because they both addressed the topic of process modelling unit and we believe that a joint response would be more effective and concise.

As stated in L175-178, a first set of processes is simulated at the ERA5-Land grid scale (0.1° resolution), specifically those involving the processing of meteorological inputs that then determine liquid inflow into soil storages. These liquid inputs, as well as evapotranspiration demand, are then averaged at the sub-basin scale since the second set of processes is modelled with a lumped approach. The first set includes rainfall-snowfall separation, snowpack evolution, and snow and ice melting (L177), while the second set consists of the processes of surface and subsurface runoff generation (L175-176), and following convolution and propagation. It should be noted that in Section S1 of the Supplement, dedicated to the model description and implementation, the general structure is represented, and reported here.

We will integrate the sentence in L178 to make it more explicit: "Ice melting is simulated only on grid points classified as having afferent glaciers according to RGI outlines". This matching results in a percentage of glacierized area at the sub-basin scale, which is considered static in all scenarios except the 1D in which it evolves according to a volume-area scaling approach. While the matching between ERA5-Land and RGI does not vary over time, the value of the glacierized fraction can change dynamically only in Scenario 1D. Finally, since ice melting here represents the contribution from water already stored in the glaciers, it is simulated only in the presence of the latter.

The coarse resolution at which some key processes are modelled, although not deviating from similar applications in the literature (L171-173), is a known critical factor of the study, discussed in detail in the Section 4.4 "Current limitations".

[Figure]

**General structure of the model described in this study**

Line 190: How are the initial conditions defined for standard grid points and for glacier grid points?

The hydrological model has three state variables expressed in terms of water storages (see also model structure represented in the Supplement and reported in the previous response). The snowpack storage was assumed null at the start of the model warmup, while the upper (layer 1) and lower (layer 2) soil storages were both half of their total thickness (i.e., $W_{max,1}$ and $W_{max,2}$, respectively). These initial conditions will be specified in Section S1 of the Supplement, which focuses on the model description and implementation. It should be noted, however, that the length of the warmup (2 water years, from June 1998 to May 2000) was set such as to minimize the dependence of the state variables on the arbitrary initial condition. In this sense, it was also necessary to avoid the potential compensation for

underestimated precipitation through depletion of initial storage capacities. Although there was no in-depth analysis of the effect of initial conditions, the established values have been verified compatible with the study's configuration and objectives. In this perspective, below are some example graphs, representing fluxes and states for a headwater sub-basin that includes glaciers during the study period.

As stated in L175-178, at the sub-basin scale the soil water balance module was implemented in a lumped way, while the snow and glacier modules operated on the ERA5-Land grid points (which may or may not have afferent glaciers). Ice melt is not controlled by state variables additional to snowpack storage. In the case of the dynamic glacier module, the contributing glacierized area was initialized according to the RGI dataset, which refers to the year 2000 and therefore adequately approximates the beginning of the study period.

[Figure]

Line 225: Could you specify which snow and glacier model parameters are used and how many there are? Which parameters are fixed (and at what values), and which ones are calibrated?

There are 6 parameters in snow and glacier modules, 4 of which are fixed a priori and 2 are calibrated. Information on these was given in the main text in L206-209 and is summarized here in the following table. To improve clarity on this point, this information will also be included in a more general table referring to all the model parameters, both calibrated and not (see the reply to the comment on L312, where this table is showed).

| Parameter | Fixed value | Calibration range |
|---|---|---|
| $T_0$ | 0 °C | |
| $k_{ice}$ | 1.3 | |
| $Z_{thr}$ | 5000 m | |
| $scale$ | 50 m | |
| $DDF_{snow,min}$ | | $1 \div 5$ mm °C$^{-1}$ d$^{-1}$ |
| $DDF_{ratio}$ | | $1 \div 2$ |

Lines 245–250: Please define P and its units. Also, specify the parameter ranges and provide references for CF in methods adj1 and adj2, as well as for the parameters in Equation 5.

The precipitation P, which was introduced in L133, will be explicitly referred to in L246 with the indication of the unit of measurement [mm]. The parameter ranges for the various adjustment approaches will be reported later in the main text in a general table structured by processes (see the reply to the comment on L312, where this table is shown). References for the different adjustment methods will be given.

Line 253: The calibration period is three times longer than the validation period. Could you discuss or justify this choice? Reducing the calibration period could have allowed validation of the glacier module, which represents one of the novelties of this work.

As stated in L253-255, the calibration period covers 14 water years (2001-2014), which becomes 13 for streamflow data (2014 was excluded), while the validation period lasts 6 water years (2015-2020). Therefore, the calibration years are just over double the validation years, not deviating from common practices. The calibration period was set to approximately match the temporal coverage of the reference glacier loss data, calculated from the differences between two satellite-based attributed to the years 2000 (SRTM mission) and 2014 (TanDEM-X mission) respectively. Therefore, the reference data are associated with a single time window without intermediate information, implying the unavailability of a second period to validate the modelled glacier loss. Finally, the end of the validation period considered (i.e., 2020) is determined by the availability of streamflow data provided to us.

Line 265: All calibration scenarios should produce the same output. In some cases these outputs result from uncalibrated parameters, while in others they come from calibrated ones. It appears that the reliability of the uncalibrated parameters is not assessed against reference data. Is there a reason for this?

Line 266: Could you specify how many parameters are involved in each scenario?

We have merged these two comments because they addressed the same main topic, namely calibrated and fixed parameters in the several scenarios, and we believe that a joint response to both would be more effective and concise. The presentation on this point was evidently not clear enough in the original submission, and the opportunity offered by the reviewer is taken to improve it and eliminate any ambiguity.

Information on the number of calibrated parameters was reported in L293-294, with an overview provided in Tab. S1 in the Supplement. As it can be seen, in the original simulations in each scenario there were the same uncalibrated parameters and largely the same calibrated parameters (only scenarios 1B and 1C differ in terms of precipitation adjustment structure, changing the number of calibrated parameters from 13 to 14). In other words, the scenarios do not differ substantially in terms of which parameters are alternatively calibrated or not. Furthermore, there are no processes in which all the parameters have been set a priori, making it difficult to attribute to these the ability or otherwise to capture the corresponding reference data.

On this point we will provide updates to consider a further scenario (Scenario 5), integrated following the comments of the first referee (see AC1). Additionally, Tab. S1 will be updated to include both calibrated and uncalibrated parameters and moved to the main text (see the reply to the comment on L312, where this new version of the table is provided).

Parameters were calibrated to reproduce reference hydrological fluxes in the basin (whose selection varies with the scenario), namely outgoing streamflow and AET, and the contribution of incoming glacier melting (additional to precipitation, which was also adjusted). We confirm that, regardless of the target fluxes, the model reproduces the same processes and outputs in every scenario, except for those (Scenarios 2-4) in which the glacier module is switched off (see reply to comment on L268). Aside from that, it is worth noting that in Scenarios 1-5 the model is structurally identical, that Scenarios 1B and 1C differ from the previous ones only for the preprocessing of incoming precipitation, and that in Scenario 1D the evolution of the glacier area is also simulated.

Line 268: Is glacier melt simulated in Scenario 2? If not, how do you justify this?

As stated in L290-293, glacier mass losses are not simulated in the absence of specific constraints (i.e., $\Phi_4$ in the objective function), mainly due to underestimation of precipitation which would tend to be compensated by overestimation of modelled ice melt, considering also that the process that is thus neglected has a limited impact on the water budget at basin scale.

Line 312: You might consider beginning the results section with a figure or table from the main text, or alternatively moving S1 to the main text if it is essential.

We will insert in the main text the updated table showing the values of the calibrated and fixed parameters, also reported below.

| Process | Parameter | Range | | Scenario | | | | | | | |
|---|---|---|---|---|---|---|---|---|---|---|---|
| | | min | max | 1 | 2 | 3 | 4 | 5 | 1B | 1C | 1D |
| Potential evapo-transpiration | $k_{PET}$ | 0.6 | 1.4 | 1.075 | 1.025 | 0.610 | *0.6* | 0.853 | 1.211 | 0.941 | 1.008 |
| Snow and ice melting | $T_0$ | - | - | [0] | [0] | [0] | [0] | [0] | [0] | [0] | [0] |
| | $k_{ice}$ | - | - | [1.3] | [1.3] | [1.3] | [1.3] | [1.3] | [1.3] | [1.3] | [1.3] |
| | $Z_{thr}$ | - | - | [5000] | [5000] | [5000] | [5000] | [5000] | [5000] | [5000] | [5000] |
| | scale | - | - | [50] | [50] | [50] | [50] | [50] | [50] | [50] | [50] |
| | $DDF_{snow,min}$ | 1 | 5 | 2.493 | 4.478 | 4.288 | 4.236 | 2.511 | 2.676 | 2.569 | 2.533 |
| | $DDF_{ratio}$ | 1 | 2 | 1.459 | 1.739 | *2* | 1.774 | 1.463 | 1.438 | 1.459 | 1.443 |

| Process | Parameter | | | | | | | | | | |
|---|---|---|---|---|---|---|---|---|---|---|---|
| Glacier area evolution | $c$ | - | - | - | - | - | - | - | - | - | [0.034] |
| | $\gamma$ | - | - | - | - | - | - | - | - | - | [1.375] |
| Streamflow propagation | $k_c$ | 2 | 30 | 6.997 | 9.598 | 29.79 | 20.95 | [16] | 23.97 | 23.52 | 25.71 |
| | $k_D$ | 1 | 15 | 13.19 | 8.515 | 3.398 | 12.00 | [8] | 11.99 | 11.83 | 8.195 |
| Surface runoff from infiltration excess | $\alpha$ | 0.5 | 10 | 3.997 | 5.715 | 8.276 | 7.307 | 2.730 | 4.355 | 4.545 | 5.242 |
| Percolation from 1st soil layer | $K_{s,1}$ | 0.1 | 200 | 104.5 | 106.2 | 146.6 | 160.8 | 1.524 | 126.2 | 53.96 | 128.0 |
| | $m_1$ | 5 | 30 | 16.13 | 17.61 | 21.72 | 22.33 | 20.15 | 12.68 | 20.42 | 19.22 |
| Percolation from 2nd soil layer | $K_{s,2}$ | 0.1 | 200 | 95.08 | 100.3 | 39.01 | 158.9 | [4.472] | 155.4 | 75.39 | 82.96 |
| | $m_2$ | 5 | 30 | 18.25 | 17.41 | 23.10 | 19.97 | [17.5] | 21.88 | 27.54 | 17.41 |
| Surface and subsurface runoff convolution | $\gamma_{GIUH}$ | 0.1 | 6 | 0.427 | 0.706 | 0.753 | 0.103 | [3.05] | 0.489 | 0.388 | 3.750 |
| | $\gamma_{LR}$ | 2 | 10 | 5.618 | 4.615 | 5.560 | 2.071 | [6] | 5.020 | 3.389 | 6.319 |
| Precipitation bias adjustment | CF | 1 | 1.4 | 1.290 | 1.325 | 1.223 | 1.214 | 1.310 | - | - | 1.280 |
| | $CF_{May-Oct}$ | 1 | 1.6 | - | - | - | - | - | 1.162 | - | - |
| | $CF_{Nov-Apr}$ | 1 | 1.6 | - | - | - | - | - | 1.532 | - | - |
| | $CF_{COE}$ | 0.6 | 1.4 | - | - | - | - | - | - | 1.103 | - |
| | $CF_{EXP}$ | 0.5 | 1.5 | - | - | - | - | - | - | 0.690 | - |

**Table 2. Model parameters and calibrated and fixed values for the different scenarios. In italic the calibrated values equal to the upper or lower bound of the parameter range, while the uncalibrated parameters are reported in square brackets. Information on parameters not described in the main text can be found in Section S1 of the Supplement. $DDF_{snow,min}$ is reported in mm °C$^{-1}$ d$^{-1}$, $T_0$ in °C, $Z_{thr}$ and scale in m, $c$ in km$^{3-2\gamma}$, $K_{s,1}$ and $K_{s,2}$ in mm h$^{-1}$. $DDF_{ratio}$ indicates the ratio of $DDF_{snow,max}$ to $DDF_{snow,min}$. The soil storage depths $W_{max,1}$ and $W_{max,2}$, which are cross-cutting parameters in the processes related to surface and subsurface runoff generation, are set at 500 and 3000 mm respectively.**

Line 423: Regarding the term "lumped," does this refer to the spatial basin average? Could you provide a spatially distributed quantitative metric (e.g., r, $R^2$, or bias) to assess how modelled and reference AET compare across the basin?

The reviewer's interpretation of the term "lumped" is correct, but we will replace it with the less ambiguous wording of "basin-wide averaged AET". Fig. 5c will report the sub-basin scale distribution of the differences in mm/year between the mean AET of MISDc-2L and GLEAM v4 (see the reply to the comment on L145).